# BEYOND MANUALS AND TASKS: INSTANCE-LEVEL CONTEXT LEARNING FOR LLM AGENTS

## ABSTRACT

Large language model (LLM) agents typically receive two kinds of context: (i) environment-level manuals that define interaction interfaces and global rules, and (ii) task-level guidance or demonstrations tied to specific goals. In this work, we identify a crucial but overlooked third type of context, **instance-level context**, which consists of verifiable and reusable facts tied to a specific environment instance, such as object locations, crafting recipes, and local rules. We argue that the absence of instance-level context is a common source of failure for LLM agents in complex tasks, as success often depends not only on reasoning over global rules or task prompts but also on making decisions based on precise and persistent facts. Acquiring such context requires more than memorization: the challenge lies in efficiently exploring, validating, and formatting these facts under tight interaction budgets. We formalize this problem as Instance-Level Context Learning (ILCL) and introduce AutoContext, our task-agnostic method to solve it. AutoContext performs a guided exploration, using a compact *TODO forest* to intelligently prioritize its next actions and a lightweight plan–act–extract loop to execute them. This process automatically produces a high-precision context document that is reusable across many downstream tasks and agents, thereby amortizing the initial exploration cost. Experiments across TEXTWORLD, ALFWORLD, and CRAFTER demonstrate consistent gains in both success and efficiency: for instance, ReAct's mean success rate in TEXTWORLD rises from 37% to 95%, while IGE improves from 81% to 95%. By transforming one-off exploration into persistent, reusable knowledge, AutoContext complements existing contexts to enable more reliable and efficient LLM agents. Our code is available at `https://anonymous.4open.science/r/context_learning_anonymized-3043`

## 1 INTRODUCTION

Large language model (LLM) agents are increasingly deployed in interactive, partially observable environments where they must act, observe, and adapt over extended horizons. As the full state is never directly accessible, an agent must construct beliefs from streaming observations, and continually revise its plan to reach to achieve the intended objective. For example, in a household domain, an agent may need to navigate through rooms, collect ingredients, and finally, use the proper tools to cook a meal. To support such decision making process, existing approaches (e.g., Chen et al., 2024; Wang et al., 2024a; Fu et al., 2024; Zhu et al., 2025) provide two primary forms of auxiliary context, illustrated in Figure 1. The first is *environment-level context*, which specifies global mechanisms and action interfaces common to all *environment instances* of a domain. The second is *task-level context*, which provides guidance specific to a target objective, including demonstrations, hints, or curricula.

However, this two-way split leaves a critical gap at deployment. When confronted with a concrete environment instance, failure often arises not from a lack of domain manuals or task instructions, but from a lack of *instance-level context*: concrete, validated facts that hold only in the current instance and cannot be deduced from manuals or task specifications (Figure 1, bottom). This includes the positions of objects, the recipes admissible in the current instance, or local rules that differ subtly across instances. Without this instance-dependent knowledge, the agent must discover basic facts before addressing the actual task, incurring unnecessary exploration cost and lowering reliability.

The inefficiencies are particularly severe in multi-task and multi-agent settings. First, every agent must independently develop exploration strategies before solving tasks. Given the diversity of agent architectures, designing effective exploration procedures for each is both costly and brittle. Second, even when the same agent returns to the same instance, it often repeats the same discovery process. Instance-dependent findings are rarely recorded in a durable form that other runs can reuse. The result is wasted interaction budget, longer trajectories, and diminished success rates.

To address this problem, we introduce *instance-level context learning*: given a previously unseen environment instance, the goal is to perform a compact, one-off exploration and distill the findings into a durable, agent-readable document $D_e$. This document records reusable, task-agnostic facts that are specific to the instance, providing a general foundation that complements both environment- and task-level context while benefiting downstream agents across diverse tasks. This new paradigm raises three intertwined challenges. **Coverage**: the document must capture general-purpose facts relevant to many future tasks. This is challenging because those facts are often hidden behind specific preconditions that require strategic exploration to uncover. **Efficiency**: exploration should be compact and avoid exponential blowup from naively enumerating trajectories, which not only wastes computation but also quickly exhausts the limited context window, leaving the extracted context unus-

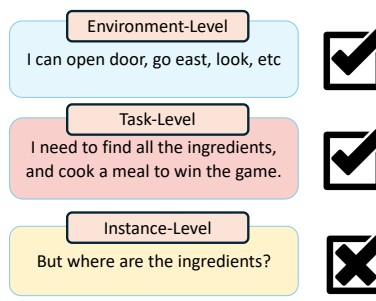

Figure 1: Three types of context. Instance-level context is usually neglected in existing methods.

able in practice. **Reliability**: extracted context should be validated and maintained as a durable document, minimizing hallucinations and brittleness when the context is later reused.

We present **AutoContext**, a task-agnostic method for instance-level context learning. It consists of three components: a Planner that identifies knowledge gaps and generates new *TODOs*, an Actor that executes these TODOs through targeted exploration, and an Extractor that validates new facts against trajectory evidence. The entire process is organized by a novel *TODO forest*, which structures the exploration and systematically exposes knowledge gaps. With this design, AutoContext achieves **coverage** by driving exploration toward informative states, ensures **efficiency** through knowledge-gap–guided exploration organized by the TODO forest, and guarantees **reliability** by validating knowledge before committing it to a durable document. Experiments across TEXTWORLD, ALF-WORLD, and CRAFTER demonstrate consistent gains: in TEXTWORLD, for example, the success rate of a ReAct agent rises from 37% to 95%, while the state-of-the-art IGE improves from 81% to 95%.

**Contributions.** Our work makes three contributions:

- We formalize the problem of *Instance-Level Context Learning (ILCL)*, establishing the objective of constructing reusable documents $D_e$ for previously unseen environment instances.
- We propose **AutoContext**, a task-agnostic method that employs a structured *TODO forest* with a *plan–act–extract* exploration loop to automatically produce validated instance-level context.
- We demonstrate through extensive experiments on TEXTWORLD, ALFWORLD, and CRAFTER that AutoContext substantially improves downstream agents: even a simple ReAct agent equipped with $D_e$ achieves performance on par with state-of-the-art methods, while stronger baselines attain further gains in both efficiency and success rates.

## 2 RELATED WORK

**Task-level Knowledge Learning.** A substantial line of research (Wang et al., 2024a; Zhang et al.; Guan et al., 2024; Zhu et al., 2025; Qiao et al., 2024; Chen et al., 2024; Zhao et al., 2024a; Fu et al., 2024; Basavatia et al., 2024; Kirk et al., 2024; Shinn et al., 2023; Chen et al., 2025; Xia et al., 2025; Wu et al., 2023) investigates how agents acquire task-specific rules. AutoManual (Chen et al., 2024) allows agents to autonomously induce environment action rules for predefined tasks via interactive trial-and-error, while ExpeL (Zhao et al., 2024a) distills both successful and failed experiences into reusable natural-language insights. Unlike these methods, which either capture global

environment mechanics or task-specific heuristics, our approach learns a persistent document of instance-dependent facts not entailed by the environment manual. This complementary knowledge can be reused by any agent across multiple tasks within the same instance.

**LLM-based Exploration.** Another line of work (Lu et al., 2025; Golchha et al., 2024; Song et al., 2024; Fang et al., 2025; Du et al., 2023) studies how LLM agents can improve exploration to uncover useful information or behaviors. Intelligent Go-Explore (IGE) (Lu et al., 2025) leverages the LLM's internalized knowledge to archive promising states and resume exploration from them. Language Guided Exploration (LGE) (Golchha et al., 2024) uses LLMs to propose promising next actions. These methods enhance task performance through improved exploration strategies, but they do not yield lasting knowledge about the environment instance.

**Instance Memory.** Other approaches (Gao et al., 2025; Holt et al.; Kagaya et al., 2024; Huang et al., 2024; Ammanabrolu & Hausknecht, 2020) augment agents with a memory or knowledge base that records useful information gathered during task execution. For example, RAP (Kagaya et al., 2024) enables LLM agents to capture and retrieve past observations via a knowledge graph, while LWM-Planner (Holt et al.) incrementally accumulates atomic facts and exploits them for improved planning through lookahead search. These approaches retain instance-dependent facts, but their exploration remains task-driven: the acquired knowledge is partial and biased toward specific tasks, limiting comprehensive coverage of the underlying instance. In contrast, our work constructs a reusable and comprehensive instance-level context that transcends individual tasks.

Due to space constraints, additional related work is provided in Appendix E.

## 3 PROBLEM FORMULATION

Let $\mathcal{E}$ denote an environment class. Each *instance* $e \in \mathcal{E}$ is modeled as a partially observable Markov decision process (POMDP)

$$e = \langle \mathcal{S}, \mathcal{A}, \mathcal{O}, T \rangle,$$

where $\mathcal{S}$ is the state space, $\mathcal{A}$ the action space, $\mathcal{O}$ the observation space (rendered to text for LLMs), and $T : \mathcal{S} \times \mathcal{A} \to \mathcal{S}$ the transition dynamics. While a standard POMDP specifies a reward function, here we deliberately decouple exploration from downstream evaluation: rather than committing to a single reward, we assume a set $\mathcal{T}_e$ of downstream tasks that may later be issued on instance $e$. Each task $t \in \mathcal{T}_e$ is drawn from a task distribution $P_e$ and paired with an LLM-based solving policy $\pi_t$. We write $U_t(\cdot)$ for the task-specific utility (e.g., success indicator, cumulative reward, etc.), evaluated on the same instance.

We formalize **Instance-Level Context Learning (ILCL)** as choosing a reusable text string $D_e$ via a one-time exploration on instance $e$ to improve future task performance. Let $\pi_{t|D_e}$ denote running the same LLM-based solver $\pi_t$ with access to $D_e$ (e.g., by conditioning its prompt on the document). The ILCL objective is

$$\max_{D_e} \mathbb{E}_{t \sim P_e} \left[ U_t\left(\pi_{t|D_e}\right) \right]. \tag{1}$$

This objective captures amortization: a single instance-level context document $D_e$, constructed once before downstream use, should raise expected utility across many future tasks and agents on the same instance. For brevity, we also refer to $D_e$ as the instance context.

Directly optimizing equation 1 requires access to the full distribution of tasks and solvers, which is intractable in practice. To address this, we introduce a *document schema* $S$ that prescribes the structure of the instance context. Formally, $S$ is an attributed entity–relation schema that abstracts the environment class, specifying the entity types, relations, and attributes to be recorded (e.g., objects, locations, preconditions). The objective of ILCL is thus to populate $D_e$ under $S$, maximizing the coverage of instance-dependent information. We describe schema design in detail in Section 4.1.

## 4 METHOD

We introduce **AutoContext**, a task-agnostic framework for ILCL. Before any downstream task is issued, AutoContext performs a compact, one-off exploration on a previously unseen environment

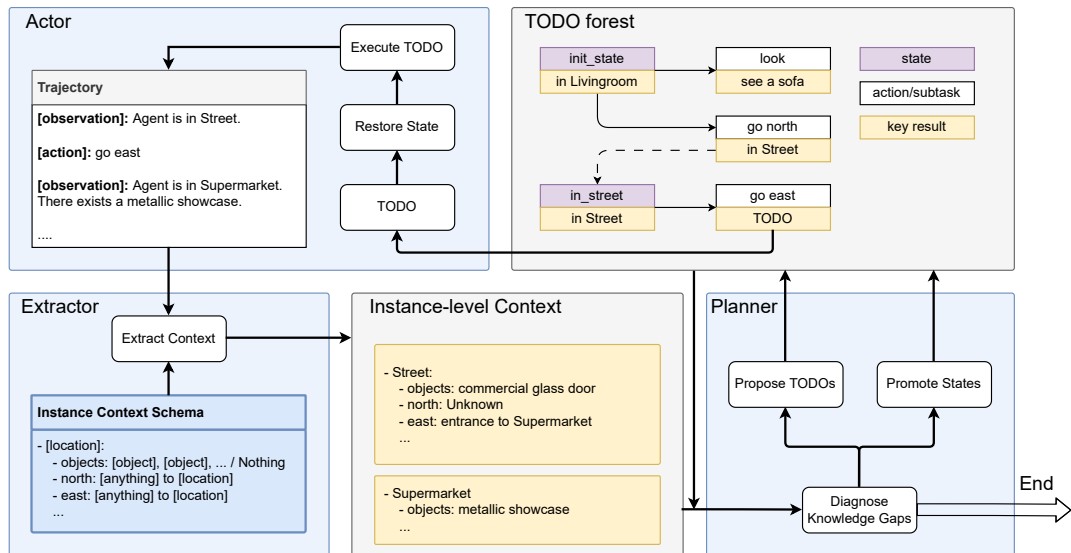

Figure 2: **Overview of AutoContext.** The **Planner** uses the current Instance-level Context and *TODO forest* to propose targeted actions (TODOs). The **Actor** executes these actions, generating a trajectory of its experience. Finally, the **Extractor** validates the information in the trajectory against a schema to update and expand the context document. This cycle iteratively builds a comprehensive and reliable summary of the environment instance.

instance $e$ to construct a reusable, agent-readable instance context $D_e$. This document captures extracted facts such as objects, locations, preconditions, and effective actions that hold in $e$. By conditioning on $D_e$, heterogeneous agents can improve both success rates and efficiency across tasks, and thus amortize the upfront exploration cost.

As illustrated in Figure 2, AutoContext is composed of two main components. The *TODO forest* provides a compact exploration representation, organizing states and subtasks into shallow trees that encourage reuse and maintain readability. The *plan–act–extract loop* iteratively propose and execute TODO nodes to perform exploration, and populate the instance context based on the trajectories, while adhering to the document schema $S$. The loop continues until the instance context is fully constructed or the exploration budget is exhausted. This produces a compact instance context $D_e$ that complements environment-level manuals and task-level guidance. We next elaborate on each component in detail.

## 4.1 INSTANCE CONTEXT SCHEMA

LLM-based agents often depend on brittle, ad-hoc prompt engineering that requires significant human effort and fails to generalize across agent architectures. To address this, we propose a principled, schema-based representation of the environment. This approach defines the structure of knowledge once from the environment's perspective, creating a durable foundation that is reusable across all tasks and agents.

Formally, we define a schema $S$ as an attributed entity-relation structure. For any instance $e$, Auto-Context constructs an instance document $D_e$ conforming to $S$. Each $D_e$ functions as a lightweight knowledge graph where nodes are typed entities (e.g., rooms, objects), attributes capture their properties, and edges encode relations (see Figure 3).

A key feature of our schema is the explicit use of `Unknown` markers for attributes that have not yet been observed. This transforms the document from a static record into a dynamic blueprint for exploration. These markers create explicit knowledge gaps (e.g., `Street: east: Unknown`) that the Planner can target. As AutoContext runs, it iteratively replaces these markers with validated facts, turning ignorance into knowledge. This design yields a reusable, agent-readable document, amortizing the one-time schema design cost across all future interactions.

```
- [location]:                              - Street:
    - objects: [object], [object], ... / Nothing    - objects: commercial glass door
    - north: [anything] to [location]          - north: Unknown
    - east: [anything] to [location]           - east: entrance to Supermarket
    ...                                        ...

- [object]:                                - cabinet_2
    - has_in_or_on: [object], [object], ... / Nothing   - has_in_or_on: fork_3, plate_1, saltshaker_1

- action: [action_name]                    - action: Make Stone Pickaxe
    - requirements: [conditions that must be met]   - requirements: Has at least 1 wood and 1 stone
    - key_result: [expected outcome]                 and faces table
    - note: [additional remarks]           - key_result: Crafts stone_pickaxe ....
                                           - note: None.

        Schema                                     Document Entries
```

Figure 3: Example schema and document entries.

## 4.2 TODO FOREST

To structure and guide exploration, we introduce the *TODO forest*, a novel data structure illustrated in Figure 2. The forest consists of multiple shallow *TODO trees*, each rooted at a *state* annotated with a succinct *state summary*. The forest is a collection of shallow *TODO trees*, each rooted in a key environment *state* and annotated with a *succinct summary*. A state corresponds to a snapshot of the environment. By promoting important nodes encountered during exploration to become new state roots, we keep the trees shallow, manage complexity, and enable informed planning by an LLM agent. The forest adapts to environmental complexity through two operational modes:

**Action Mode.** In simpler environments, each non-node represents a primitive action paired with a *key result*. The key result is a compact abstraction of the observation induced by executing the action. Unexplored nodes are marked TODO. This mode captures both successful outcomes and negative feedback (e.g., unsatisfied preconditions, syntax errors), providing a fine-grained map of the local action space.

**Agent Mode.** Agent mode is designed for complex environments with long horizons and extensive observations, where naively storing all trajectories for direct LLM inspection is infeasible. In this mode, non-root nodes correspond to higher-level subtasks, represented as agent("task description"). Control is delegated to a sub-agent (e.g., a ReAct-style agent) to execute the subtask. The sub-agent's full trajectory is stored, but its outcome is summarized via an LLM into a concise key result. This provides a hierarchical abstraction, avoiding the infeasibility of processing long, raw trajectories.

**Exploration representation and In-Context Examples.** The TODO forest compactly records the entirety of exploration. AutoContext uses this structure to resume exploration from any TODO node by replaying the trajectory from the initial state to that node, thus enabling the discovery of facts that rely on preconditions. Moreover, since the forest records both successful and failed trajectories, it simultaneously serves as a collection of in-context examples: failed attempts guide the agent away from unproductive branches, while successful ones provide reusable execution patterns.

## 4.3 PLAN-ACT-EXTRACT LOOP

AutoContext builds the instance context through a *plan–act–extract* loop driven by the TODO forest. Each iteration includes three LLM-driven pipelines: (1) the *Planner*, which expands the forest by proposing new TODOs and promoting states; (2) the *Actor*, which executes TODOs either directly (action mode) or via a delegated ReAct agent (agent mode); and (3) the *Extractor*, which updates the instance context based on the trajectories and the instance context schema. The loop continues until the instance context achieves sufficient coverage, or the exploration budget is exhausted. Prompt templates are provided in Appendix D.

**Planner: Proposing TODOs.**  The Planner is prompted to identify knowledge gaps by scanning attributes marked as `Unknown` and analyzing the TODO forest to find opportunities where continued exploration can uncover new information. For example, if the instance context indicates that the north of `Street` is `Unknown` and the forest shows that the destination remains unexplored, both signals the need to resolve this knowledge gap. The Planner then proposes candidate TODOs, which are validated against the forest. If a proposed TODO path is redundant or originates from a non-existent state, feedback is provided for regeneration. This ensures that only valid TODOs are admitted.

**Actor: Completing TODOs.**  The Actor executes the proposed TODOs and returns the resulting trajectories. Specifically, it first resumes the snapshot of the explored part of the TODO paths by replaying stored trajectories, and then executes the remaining new actions (action mode), or invokes a ReAct agent to complete the subtask (agent mode). When the environment supports save and restore, we can also store a checkpoint at the node to enable direct resumption.

**Extractor: Update Document.**  The Extractor updates the instance context based on the instance context schema and the trajectories returned by the Actor. First, it is prompted to propose a list of candidate edits with three modification types: add, update, or remove. Second, those edits for the instance context will be checked one by one, each against the trajectories and the schema. Each edit can be accepted, revised, or rejected. Finally, the Extractor is prompted to apply the accepted and revised edits to the instance context. This separation of proposing, verifying, and applying edits allows the LLM to reason about each modification in isolation, and thus ensures that the resulting instance context is both schema-compliant and of high precision.

**Planner: Proposing States.**  After updating the instance context, the Planner revisits the forest to promote selected TODO nodes into new states. Similar to proposing TODOs, the promotion is guided by the principle of resolving knowledge gaps: nodes that can uncover novel information are prioritized. The promoted states are also required to be fundamentally distinct from all existing ones, to encourage wide coverage of the instance context and avoid redundant exploration.

**Loop Control.**  At the end of each iteration, the Planner is prompted to analyze both the TODO forest and the instance context. The loop continues if knowledge gaps remain and the iteration budget is not exhausted. Otherwise exploration terminates and the finalized instance context is provided to downstream tasks as an input.

## 5 EXPERIMENTAL EVALUATION

Our experiments aim to answer the following main research questions. More experiments on contributions of different instance context schema are deferred to Appendix B.

- **RQ1:** How much performance gain can be achieved by AutoContext?
- **RQ2:** How efficient are AutoContext and the baselines with the instance context?
- **RQ3:** What is the contribution of each component of AutoContext?

### 5.1 EXPERIMENT SETUP

**Benchmarks.**  Following Lu et al. (2025), we use the TextWorld cooking benchmark with 25 randomly generated environment instances. This benchmark is deliberately challenging: agents must navigate up to 12 rooms, identify recipes, tools, and ingredients, execute multi-step preparation, and finally consume the meal to succeed. ALFWorld (Chen et al., 2024) provides 134 unseen test household environments for embodied task completion. Crafter (Hafner, 2022) is a $64 \times 64$ open-ended survival world with a technology tree, where the agent must gather resources, satisfy preconditions, and unlock advanced actions such as crafting stone tools and mining iron. For TextWorld and ALF-World, we apply AutoContext in action mode, while for Crafter we adopt agent mode to better handle its long horizons and rich observations.

Table 1: Success rates (%) on TEXTWORLD under increasing step budgets.

| Model | Method | 50 | 100 | 400 | 1600 | unlimited |
|-------|--------|-----|-----|-----|------|-----------|
| DeepSeek-V3 | ReAct | $16 \pm 7$ | $35 \pm 7$ | $37 \pm 5$ | $37 \pm 5$ | $37 \pm 5$ |
| | ReAct + AutoContext | $\mathbf{78 \pm 6}$ | $95 \pm 2$ | $95 \pm 2$ | $95 \pm 2$ | $95 \pm 2$ |
| | Reflexion | $16 \pm 11$ | $31 \pm 3$ | $45 \pm 4$ | $45 \pm 4$ | $45 \pm 4$ |
| | Reflexion + AutoContext | $73 \pm 5$ | $\mathbf{96 \pm 3}$ | $\mathbf{99 \pm 2}$ | $\mathbf{99 \pm 2}$ | $\mathbf{99 \pm 2}$ |
| | IGE | $0 \pm 0$ | $0 \pm 0$ | $36 \pm 6$ | $79 \pm 7$ | $81 \pm 8$ |
| | IGE + AutoContext | $0 \pm 0$ | $0 \pm 0$ | $72 \pm 3$ | $95 \pm 2$ | $95 \pm 2$ |
| GPT4.1 | ReAct | $33 \pm 7$ | $81 \pm 2$ | $91 \pm 8$ | $93 \pm 7$ | $93 \pm 7$ |
| | ReAct + AutoContext | $\mathbf{83 \pm 5}$ | $\mathbf{100 \pm 0}$ | $\mathbf{100 \pm 0}$ | $\mathbf{100 \pm 0}$ | $\mathbf{100 \pm 0}$ |
| | Reflexion | $35 \pm 5$ | $77 \pm 3$ | $\mathbf{100 \pm 0}$ | $\mathbf{100 \pm 0}$ | $\mathbf{100 \pm 0}$ |
| | Reflexion + AutoContext | $\mathbf{83 \pm 2}$ | $\mathbf{100 \pm 0}$ | $\mathbf{100 \pm 0}$ | $\mathbf{100 \pm 0}$ | $\mathbf{100 \pm 0}$ |
| | IGE | $0 \pm 0$ | $0 \pm 0$ | $43 \pm 12$ | $69 \pm 8$ | $72 \pm 7$ |
| | IGE + AutoContext | $0 \pm 0$ | $0 \pm 0$ | $81 \pm 2$ | $\mathbf{100 \pm 0}$ | $\mathbf{100 \pm 0}$ |

Table 2: Success rates (%) on ALFWORLD under increasing step budgets.

| Method | 5 | 10 | 40 | 160 | unlimited |
|--------|-----|-----|-----|-----|-----------|
| ReAct | $13.1 \pm 1.1$ | $48.3 \pm 3.6$ | $76.5 \pm 1.0$ | $77.5 \pm 0.8$ | $77.5 \pm 0.8$ |
| ReAct + AutoContext | $26.7 \pm 1.5$ | $\mathbf{97.3 \pm 1.7}$ | $98.0 \pm 1.1$ | $98.0 \pm 1.1$ | $98.0 \pm 1.1$ |
| IGE | $10.4 \pm 0.0$ | $10.6 \pm 0.4$ | $54.6 \pm 1.1$ | $86.9 \pm 1.9$ | $94.3 \pm 1.1$ |
| IGE + AutoContext | $\mathbf{26.9 \pm 0.4}$ | $27.2 \pm 0.4$ | $96.8 \pm 1.1$ | $98.8 \pm 0.9$ | $99.3 \pm 0.0$ |
| AutoManual | $3.4 \pm 0.4$ | $28.8 \pm 0.7$ | $94.8 \pm 0.5$ | $97.9 \pm 0.5$ | $97.9 \pm 0.5$ |
| AutoManual + AutoContext | $17.6 \pm 1.1$ | $83.2 \pm 1.2$ | $\mathbf{99.7 \pm 0.4}$ | $\mathbf{99.7 \pm 0.4}$ | $\mathbf{99.7 \pm 0.4}$ |

**Baselines.** We use the current SOTA methods as baselines and augment them with AutoContext. By construction, the instance context produced by AutoContext is both task-agnostic and agent-agnostic, and is simply appended to the prompts of baselines without any customization. The baselines are: (1) Intelligent-Go-Explore (IGE) (Lu et al., 2025): a LLM-driven go-explore methods, which search the whole environments by archiving interesting states and continue exploration from those states. (2) ReAct (Yao et al., 2023b): A classic Chain-of-Thought method, which prompts LLM to generate reasoning process. (3) AutoManual (Chen et al., 2024): A task-level context generation method, with advanced architecture for completing tasks by LLM-generated code. (4) Reflexion (Shinn et al., 2023): A method that generates self-reflections from previous trials to improve future decisions. We limit Reflexion to a maximum of three trials.

**LLMs** Unless noted, all methods are driven by DeepSeek-V3 (i.e., `DeepSeek-V3-0324` (Liu et al., 2024)), except that for CRAFTER we employ `DeepSeek-R1-0528` with AutoContext to construct instance contexts. While R1 and V3 are contemporaneous, R1 uses inference-time scaling to improve reasoning reliability. This setup allows us to investigate how lightweight models can be leveraged for downstream evaluation tasks, while more computationally intensive models are reserved for generating high-quality instance contexts. We also employ GPT-4.1 (OpenAI, 2025a) to assess the robustness of our methods across different LLMs.

## 5.2 EXPERIMENTAL RESULTS

**RQ1: How much performance gain can be achieved by AutoContext?** We first examine whether augmenting existing baselines with AutoContext yields significant improvements across benchmarks. For TextWorld and ALFWorld, we report the success rates of all methods under varying environment step limits (including the unlimited-step setting, where no constraints are imposed). For Crafter, we follow the benchmark's official evaluation protocol, reporting scores computed as the geometric mean of achievement completion rates across all runs.

TEXTWORLD. Table 1 reports the mean success rates under varying step budgets over three runs. ReAct struggles even with unlimited steps, achieving only 37% success due to frequent navigation

errors. When augmented with the instance context from AutoContext, its success rate under a 50-step budget improves from 16% to 78%, and further reaches 95% under unlimited budget. A similar performance gain can be observed for IGE. While IGE, as a search-oriented method, generally outperforms ReAct under large step budgets, its performance remains constrained by limited memory. It must discard states during exploration, leading to information loss, redundant revisits, and ultimately incomplete coverage. AutoContext resolves this by maintaining a compact TODO forest of all explored trajectories and marking unknowns to highlight knowledge gaps. Consequently, IGE improves from 36% to 72% under a 400-step budget, and from 81% to 95% with unlimited steps. These results show that the instance context is not merely an efficiency booster, but a structural solution to the inherent context limitations of LLMs, enabling exhaustive exploration where naive reasoning is insufficient.

ALFWORLD. Consistent with TextWorld, AutoContext delivers substantial improvements across all baselines (Table 2). ReAct's success rate rises from 48.3% to 97.3% and AutoManual rises from 28.8% to 83.2% under 10 steps. IGE rises from 54.6% to 96.8% under 40 steps. The largest improvement for ReAct appears at very small budgets (10 steps), where ReAct + AutoContext achieves a nearly optimal success rate of 97.3%. This demonstrates that once instance context is resolved into a structured, compact form, agents can execute complex embodied tasks with near-optimal efficiency.

CRAFTER. The official scores over two runs of all methods are presented in Figure 4. AutoContext yields marked improvements for both ReAct and Reflexion, demonstrating its generality and effectiveness in open-ended survival environments. The gain arises because AutoContext can explicitly capture preconditions and action dependencies, enabling the agent to reason about complex achievements and rules. ReAct and Reflexion then leverage this structured context to plan and execute longer-horizon strategies that naive exploration cannot sustain. Additional details on the Crafter experiments are deferred to Appendix B, due to space limitations.

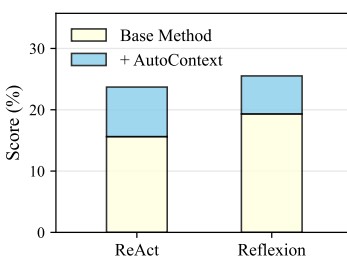

Figure 4: Scores on Crafter

**RQ2: How efficient are AutoContext and the baselines with the instance context?** We investigate the efficiency of AutoContext in constructing high-quality instance contexts, measured in terms of environment steps. Specifically, we quantify the coverage of locations and objects captured in the instance context. As shown in Figure 5, AutoContext rapidly attains over 95% coverage in TextWorld within 200 steps. For comparison, IGE achieves only about 15% success rate at the same step budget, indicating that our approach can extract nearly complete contextual information while state-of-the-art baselines remain far from effective. A similar trend is observed in ALFWorld. AutoContext requires approximately 120 steps to cover more than 95% of the objects, whereas IGE typically needs over 160 steps before its success rate stabilizes. These results consistently demonstrate the superior efficiency of AutoContext across distinct environments. Nevertheless, in both environments, IGE augmented with AutoContext also converges faster than IGE, as the instance context provides effective guidance.

To further show the efficiency of baselines augmented with AutoContext, we report the average number of steps required for successful runs in Table 3. AutoContext reduces step requirements by a significant margin and improves the efficiency of all baselines. Beyond efficiency gains, AutoContext also improves overall success rates, as established in RQ1. Taken together, these results indicate that AutoContext not only accelerates convergence but also enhances the effectiveness of the baselines.

**RQ3: What is the contribution of each component of AutoContext?** We conduct ablation studies to assess the contribution of each component in AutoContext by systematically removing or replacing them, and measuring the resulting performance when combined with ReAct. The results are reported in Table 4. Specifically, we consider three variants: (i) *w/o TODO Forest*, where the Planner generates TODO paths based only on the current instance context and recent trajectory, without access to prior successful or failed attempts. (ii) *w/o Planner*, where the Planner is replaced with random actions, removing targeted exploration. and (iii) *w/o Extractor*, where the Extractor is replaced with a naive LLM prompt that updates the instance context without adhering to the schema.

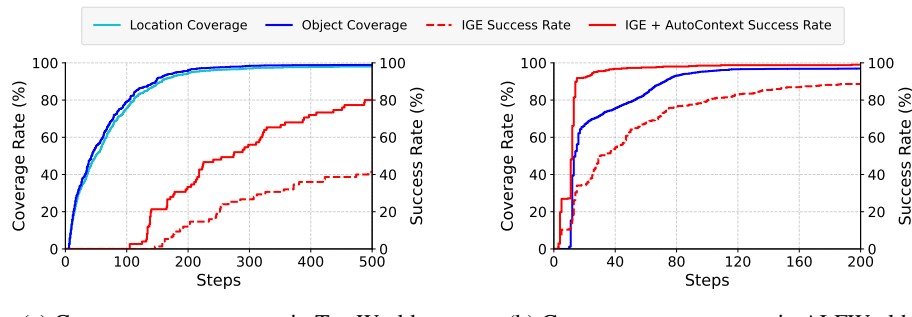

(a) Coverage rates vs. steps in TextWorld      (b) Coverage rates vs. steps in ALFWorld

Figure 5: Converge rates and success rates across different environments. With AutoContext, coverage and success rates rise rapidly as the number of steps increases.

Table 3: Average steps of successful runs of all methods.

| Method | TextWorld | | ALFWorld | |
|---|---|---|---|---|
| | Baseline | + AutoContext | Baseline | + AutoContext |
| ReAct | 60.7 | **42.7** | 11.4 | **6.6** |
| Reflexion | 87.2 | **48.5** | - | - |
| IGE | 594.5 | **320.5** | 60.6 | **13.4** |
| AutoManual | - | - | 17.4 | **8.4** |

Table 4: Ablation study of AutoContext components.

| Method | TextWorld | ALFWorld |
|---|---|---|
| ReAct + AutoContext (Ours) | $95 \pm 2$ | $98.5 \pm 0.7$ |
| Ours w/o TODO forest | $51 \pm 5$ | $94.7 \pm 3.1$ |
| Ours w/o Planner | $40 \pm 3$ | $89.3 \pm 2.3$ |
| Ours w/o Extractor | $81 \pm 4$ | $81.8 \pm 1.5$ |
| ReAct | $37 \pm 5$ | $77.5 \pm 0.8$ |

The ablations expose distinct failure modes. Removing the TODO Forest leads to pronounced degradation in TextWorld, as it requires more in-context examples to navigate such a long-horizon environment effectively. Without the Planner, exploration becomes unguided, and the agent fails to identify informative trajectories, resulting in a sharp performance collapse. Finally, replacing the Extractor has a strong impact in ALFWorld. Since this environment generates a large volume of irrelevant observations, the absence of schema guidance causes the Extractor to record verbose and uninformative content, making it harder to capture important information and hindering task completion.

## 6 CONCLUSION AND LIMITATIONS

This paper introduces AutoContext, a principled framework for instance context learning that converts systematic exploration into reusable knowledge. We demonstrate that constructing a high-precision, compact, and exhaustive instance context substantially enhances the performance of downstream LLM agents, enabling more robust reasoning and efficient planning. This result highlights that future general-purpose agents may increasingly rely on structured, per-instance context as a foundational component of intelligence.

Nonetheless, our approach has limitations. Its effectiveness decreases when the number of observations exceeds the LLM's context capacity (e.g., large e-commerce catalogs), where retrieval-augmented generation remains essential. In such cases, instance context should emphasize operational structures (e.g., interface logic, navigation schema) rather than exhaustive details. In addition, the instance-context schema currently requires manual design. A promising direction for future work is to automatically induce such schemas from the environment.

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

## APPENDIX CONTENTS

## A  LLM USAGE STATEMENT

We used GPT-5 (OpenAI, 2025b) to polish the writing of this paper, including grammar correction, sentence reorganization, and generating code for plotting experimental results as figures and tables. All ideas and methods were developed by the authors, and all content and conclusions presented in this paper have been thoroughly verified by the authors.

## B  ADDITIONAL EXPERIMENTAL DETAILS

This section presents extended details complementing the main experimental results. We first introduce the schema of the instance context and illustrate its structural design with concrete examples. We then analyze the effectiveness of different schema components through controlled ablation experiments. Finally, we provide additional details for the exploration cost of AutoContext.

### B.1  INSTANCE CONTEXT SCHEMA

Our instance context is represented in a structured markdown format. The schema consists of two major components: `Observations` and `Action Rules`. Each component contains multiple entries. The `Action Rules` schema is identical across environments, which highlights its generality. The `Observations` section is environment-specific but remains intuitive and can be manually crafted with minimal customization for each domain. The `Observations` component enables AutoContext to capture factual information such as locations, objects, and their properties, while the `Action Rules` component provides the operational rules necessary for interacting with a given environment instance. We detail the concrete schemas for TextWorld, ALFWorld, and Crafter below.

> **Schema for TextWorld**
>
> ```
> #### Observations
> - [location]:
>   - objects: [object], [object], ... / Nothing
>   - west: [anything] to [location]/Unknown
> ```

```
      - east: [anything] to [location]/Unknown
      - north: [anything] to [location]/Unknown
      - south: [anything] to [location]/Unknown

   ...

   #### Action Rules
   - action: [action_name]
      - requirements: [conditions that must be met]
      - key_result: [expected outcome]
      - note: [additional remarks]

   ...
```

**Schema for ALFWorld**

```
   #### Observations
   - [object]
      - has_in_or_on: [object], [object], ... / Unknown / Nothing

   ...

   #### Action Rules
   - action: [action_name]
      - requirements: [conditions that must be met]
      - key_result: [expected outcome]
      - note: [additional remarks]

   ...
```

**Schema for Crafter**

```
   #### Observations
   - Position [x, y]: can see [object], [object], ... / Nothing
   - Position [x, y]: can see [object], [object], ... / Nothing
   ...

   #### Action Rules
   - action: [action_name]
      - requirements: [conditions that must be met]
      - key_result: [expected outcome]
      - note: [additional remarks]

   ...
```

## B.2 INSTANCE CONTEXT EXAMPLE

We present an example instance context learned by AutoContext. The example demonstrates how AutoContext systematically records both resource information and precise action rules. For observations, AutoContext explores the environment to gather resource information. For action rules, it first establishes preconditioning nodes, then conducts additional exploration with retries to identify feasible strategies to complete the actions, and ultimately derives the correct `Action Rules`.

**Example Instance Context on Crafter**

```
#### Observations

- Position [42, 36]: can see
- Position [42, 39]: can see cow, grass
```

```
- Position [44, 35]: can see tree
- Position [44, 36]: can see grass, tree
- Position [44, 37]: can see grass, path
- Position [44, 38]: can see grass, stone, path
- Position [45, 38]: can see stone, path, diamond
- Position [45, 39]: can see path, stone, diamond
- Position [44, 39]: can see path, stone, grass
- Position [44, 40]: can see stone, diamond, grass

#### Action Rules

- action: Do
  - requirements:
    - The front adjacent cell is an interactable object (grass, tree,
        stone) or a mob.
    - If the object is stone, the agent must have a pickaxe (
        wood_pickaxe, stone_pickaxe, or iron_pickaxe) in inventory.
  - key_result:
    - Collects resources from the front adjacent natural objects (
        sapling from grass, wood from tree, stone from stone). Tree/
        stone are removed. Grass remains.
    - Attacks the front adjacent mob, dealing damage. If killed after
        sufficient attacks, mob is removed and provides status benefits
        (e.g., killing a cow restores food).
  - note: Weapons optional for attacking. Mobs cause passive damage
      when nearby (e.g., zombies reduce health). Killing requires
      consecutive attacks (e.g., cow needs 3 hits). Combat may cause
      health loss near hostile mobs even without attacking. Collecting
      resources from grass may require multiple attempts.

- action: Make Stone Pickaxe
  - requirements: Agent has >= 1 wood and >= 1 stone in inventory;
      adjacent to a table
  - key_result: Consumes 1 wood and 1 stone; adds 1 stone_pickaxe to
      inventory
  - note: Crafting requires adjacency to a table in any direction (not
      necessarily front).

- action: Make Stone Sword
  - requirements: Agent has >= 1 wood and >= 1 stone in inventory;
      adjacent to a table
  - key_result: Consumes 1 wood and 1 stone; adds 1 stone_sword to
      inventory
  - note: Crafting requires adjacency to a table.

- action: Make Wood Pickaxe
  - requirements: Agent has at least one wood in inventory; agent is
      adjacent to a table
  - key_result: Consumes one wood; adds one wood_pickaxe to inventory
  - note: Crafting requires only adjacency to a table.

- action: Make Wood Sword
  - requirements: Agent has at least one wood in inventory; agent is
      adjacent to a table
  - key_result: Consumes one wood; adds one wood_sword to inventory
  - note: Crafting requires only adjacency to a table.

- action: Move East
  - requirements: The adjacent east cell is grass or path and contains
      no obstacles (e.g., stone or table).
  - key_result: Agent moves one cell east
  - note: Fails if blocked by obstacles (e.g., stone, table).
```

```
- action: Move North
  - requirements: The adjacent north cell is grass or path and contains
      no obstacles (e.g., stone or table).
  - key_result: Agent moves one cell north
  - note: Fails if blocked by obstacles (e.g., stone, table).

- action: Move South
  - requirements: The adjacent south cell is grass or path and contains
      no obstacles (e.g., stone or table).
  - key_result: Agent moves one cell south
  - note: Fails if blocked by obstacles (e.g., stone, table).

- action: Move To [x, y]
  - requirements: Agent knows the exact coordinates of the target
      position, is not currently at [x, y], and the direct path to [x,
      y] is clear (no obstacles blocking movement).
  - key_result: Agent moves toward the specified position. If the path
      is blocked, movement stops at the last valid position.
  - note: Use for distant coordinates with known positions. For nearby
      objects (within a few steps), use directional moves (e.g., Move
      East).

- action: Move West
  - requirements: The adjacent west cell is grass or path and contains
      no obstacles (e.g., stone or table).
  - key_result: Agent moves one cell west
  - note: Fails if blocked by obstacles (e.g., stone, table).

- action: Place Plant
  - requirements:
      - Agent has at least one sapling in inventory
      - The adjacent cell in the front direction is grass
  - key_result: Places a plant at the front adjacent cell (replacing
      terrain), consuming one sapling
  - note: Similar to Place Table/Stone; requires front adjacent grass

- action: Place Stone
  - requirements: Agent has at least one stone in inventory; the
      adjacent cell in the front direction is grass
  - key_result: Places a stone block at the front adjacent cell,
      replacing the terrain, and consumes one stone. The placed stone
      becomes a collectible object (e.g., via "Do" with pickaxe).
  - note: Similar to Place Table; requires front adjacent grass terrain
      . The placed stone becomes a permanent obstacle and interactable
      resource.

- action: Place Table
  - requirements: Agent has at least 2 wood in inventory; the adjacent
      cell in the front direction is grass.
  - key_result: A table is placed at the front adjacent cell replacing
      the grass, consuming 2 wood from inventory.
  - note: The table blocks movement but enables crafting of tools when
      adjacent.
```

### B.3 EXPERIMENTS ON DIFFERENT COMPONENTS OF INSTANCE CONTEXT

**Setup.** We evaluate the impact of different components of the instance context on two benchmarks, TextWorld and Crafter, by augmenting ReAct with varying sections of the context. Unlike our main experiments, in this evaluation no action illustrations or usage instructions are provided; both AutoContext and ReAct are given only the action list and must infer the correct application of each action. We examine four configurations: (i) ReAct, which operates without the generated instance context; (ii) ReAct + Observations, where the agent is provided only the observation component

Table 5: Performance on Crafter (score) and TextWorld (success) under minimal prompting. We compare ReAct augmented with different components of the instance context.

| Method | Crafter (Score) | TextWorld (Success) |
|---|---|---|
| ReAct | 15.6 | 22.7 |
| ReAct + Observations | 18.0 | 82.7 |
| ReAct + Action Rules | 22.2 | 25.3 |
| ReAct + AutoContext | **23.7** | **84.0** |

generated by AutoContext; (iii) ReAct + Action Rules, where the agent is provided only the action rules; (iv) ReAct + AutoContext, our default setting, in which both observations and action rules are included.

**Results on Context Components.** The results are shown in Table 5. In Crafter, observations only increase the score from 15.6 to 18.0, while Action Rules lead to a larger improvement to 22.2. In TextWorld, the dominant factor is Observations: the success rate rises from 22.7% to 82.7%, whereas action rules alone provide only a small gain (25.3%). This is because TextWorld relies heavily on accurate navigation, while Crafter benefits more from procedural rules that capture long-horizon dependencies. Combining both components achieves the best performance in both environments, reaching 23.7 in Crafter and 84.0% in TextWorld. This indicates that observations and action rules contribute complementary forms of knowledge, and leveraging both is crucial for robust performance across diverse domains.

**Per-Achievement Results on Crafter.** To better understand the contribution of contextual information, we report the completion rates of all achievements and compare the effects of AutoContext with observations and action rules only. As shown in Table 6, ReAct + AutoContext improves most achievements over vanilla ReAct, especially more challenging ones such as Collect Iron, Collect Stone, Defeat Skeleton, Defeat Zombie, Make Stone Pickaxe, Make Stone Sword, and Place Furnace. AutoContext successfully discovers the rules underlying these achievements, and ReAct leverages them to achieve higher scores. However, we also observe two simple achievements, Collect Sapling and Wake Up, where ReAct + AutoContext underperforms ReAct. These tasks are non-emergent and can be easily solved by the native ReAct agent through simple trial, whereas ReAct + AutoContext tends to prioritize advanced survival-oriented activities, exhibiting longer planning horizons and more structured strategies. Moreover, several highly challenging achievements, including Collect Diamond, Eat Plant, and Make Iron Sword, remain unsolved. These tasks involve numerous preconditions and demand very long survival times under specific conditions, which we leave for future exploration.

### B.4 Exploration Cost of AutoContext

Finally, we report the ... Table 7 reports the average number of environment steps required by AutoContext to construct instance context. This process corresponds to a one-time preprocessing cost per environment instance. Once generated, the instance context can be reused across multiple downstream agents and tasks, making the amortized cost negligible in practice.

## C  Case Study

We illustrate how AutoContext generates instance context using an example from TextWorld. We then compare two trajectories: (1) a ReAct agent that fails due to missing crucial ingredients; (2) a ReAct agent that leverages the instance context to successfully complete the task.

### C.1  Instance Context Generation

We present example snippets from the AutoContext log. The TODO forest is represented by indented text, with child nodes shown at deeper indentation levels. Within each node, the action and

Table 6: CRAFTER achievement success (%) under different settings

| Achievement | ReAct | ReAct + Observations | ReAct + Action Rules | ReAct + AutoContext |
|---|---|---|---|---|
| Collect Coal | 22.5% | 45.0% | 37.5% | 52.5% |
| Collect Diamond | 0.0% | 0.0% | 0.0% | 0.0% |
| Collect Drink | 75.0% | 72.5% | 80.0% | 70.0% |
| Collect Iron | 2.5% | 15.0% | 15.0% | 27.5% |
| Collect Sapling | 30.0% | 20.0% | 12.5% | 15.0% |
| Collect Stone | 35.0% | 60.0% | 57.5% | 80.0% |
| Collect Wood | 100.0% | 97.5% | 100.0% | 100.0% |
| Defeat Skeleton | 5.0% | 10.0% | 20.0% | 12.5% |
| Defeat Zombie | 42.5% | 27.5% | 70.0% | 70.0% |
| Eat Cow | 75.0% | 75.0% | 57.5% | 67.5% |
| Eat Plant | 0.0% | 0.0% | 0.0% | 0.0% |
| Make Iron Pickaxe | 0.0% | 0.0% | 2.5% | 2.5% |
| Make Iron Sword | 0.0% | 0.0% | 0.0% | 0.0% |
| Make Stone Pickaxe | 17.5% | 30.0% | 52.5% | 60.0% |
| Make Stone Sword | 15.0% | 20.0% | 35.0% | 47.5% |
| Make Wood Pickaxe | 72.5% | 67.5% | 92.5% | 95.0% |
| Make Wood Sword | 55.0% | 40.0% | 72.5% | 57.5% |
| Place Furnace | 17.5% | 20.0% | 35.0% | 32.5% |
| Place Plant | 20.0% | 12.5% | 10.0% | 15.0% |
| Place Stone | 22.5% | 40.0% | 35.0% | 42.5% |
| Place Table | 85.0% | 67.5% | 100.0% | 95.0% |
| Wake Up | 52.5% | 57.5% | 45.0% | 35.0% |

Table 7: Average environment steps required for instance context construction

| | TextWorld | ALFWorld | Crafter |
|---|---|---|---|
| Average Steps | 418.0 | 177.9 | 724.8 |

key result (or the state and its summary) are separated by a colon. Below are example snippets taken from the middle of the AutoContext log; parts of the log are omitted for brevity and clarity.

AutoContext diagnoses two knowledge gaps: the east and north of the Corridor are unknown, based on the current TODO forest and the instance context.

---

**Current TODO Forest**

```
- init_state: Agent's location: Livingroom. The livingroom contains an
    empty sofa. The livingroom has a closed fiberglass door leading
    south, an exit to the east without a door, and an exit to the north
    . Agent is hungry and needs to cook a meal.
  - examine sofa: The sofa is reliable.
    - inventory: You are carrying nothing.
  - go east: Agent's location: Bedroom. The bedroom contains a large
      empty bed. The bedroom has an entranceway to the north without a
      door and an exit to the west without a door.
  - go north: Agent's location: Kitchen. The kitchen contains a fridge,
       an oven, a table with a cookbook, a counter with a raw purple
      potato, a red apple, a raw yellow potato and a knife, and an
      empty stove. The kitchen has a closed frosted-glass door leading
      north, an exit to the east without a door, and an entranceway to
      the south without a door. [reach in_kitchen]
  - look: Agent's location: Livingroom. The livingroom contains an
      empty sofa. The livingroom has a closed fiberglass door leading
      south, an exit to the east without a door, and an exit to the
      north.
    - examine sofa: The sofa is reliable.
      - inventory: You are carrying nothing.
  - open fiberglass door: You open fiberglass door.
```

```
        - go south: Agent's location: Driveway. The driveway has an open
            fiberglass door leading north and an exit to the east without a
            door.

    - in_kitchen: Agent is in the Kitchen. The kitchen contains a fridge,
        an oven, a table with a cookbook, a counter with a raw purple
        potato, a red apple, a raw yellow potato and a knife, and an empty
        stove. The kitchen has a closed frosted-glass door leading north,
        an exit to the east without a door, and an entranceway to the south
        without a door.
        - examine cookbook: Recipe requires: orange bell pepper, pork chop,
            purple potato, red onion, white onion. Preparation steps: dice
            orange bell pepper and white onion, slice pork chop and purple
            potato and red onion, grill orange bell pepper/pork chop/purple
            potato/white onion, roast red onion, then prepare meal.
        - examine fridge: The fridge looks durable and is closed.
        - go east: Agent's location: Corridor. The corridor has a closed
            sliding patio door leading north, an exit to the east without a
            door, an entranceway to the south without a door, and an
            entranceway to the west without a door.
        - open frosted-glass door: You open frosted-glass door.
            - go north: Agent's location: Pantry. The pantry contains a wooden
                shelf. The pantry has an open frosted-glass door leading south.
        - take knife from counter: You take the knife from the counter.
            - take raw purple potato from counter: You take the purple potato
                from the counter.

    - in_street: Agent is in the Street. The street has a closed sliding
        door leading east and an exit to the west without a door.
```

**Instance Context**

```
...

- Corridor:
    - objects: Nothing
    - west: entranceway (without door) to Kitchen
    - east: exit (without door) to Unknown
    - north: closed sliding patio door to Unknown
    - south: entranceway (without door) to Bedroom

...
```

AutoContext then proposes two TODO paths to navigate the environment to find out the east and the north of Corridor. The TODO forest shows that it can arrive in Corridor first by going east from the state in_kitchen. So the two paths start with in_kitchen -> go east and then explore the east and the north.

**Proposed TODO**

```
Knowledge gaps:
- east exit from Corridor (leads to Unknown)
- north sliding patio door from Corridor (leads to Unknown)
- east sliding door from Street (leads to Unknown)

TODO:
in_kitchen -> go east -> open sliding patio door -> go north
```

**Proposed TODO**

```
Knowledge gaps:
- east exit from Corridor (leads to Unknown)

TODO:
in_kitchen -> go east -> go east
```

After the exploration, the TODO forest is updated with the results.

**Current TODO Forest**

```
- init_state: Agent's location: Livingroom. The livingroom contains an
    empty sofa. The livingroom has a closed fiberglass door leading
    south, an exit to the east without a door, and an exit to the north
    . Agent is hungry and needs to cook a meal.
  ...

- in_kitchen: Agent is in the Kitchen. The kitchen contains a fridge,
    an oven, a table with a cookbook, a counter with a raw purple
    potato, a red apple, a raw yellow potato and a knife, and an empty
    stove. The kitchen has a closed frosted-glass door leading north,
    an exit to the east without a door, and an entranceway to the south
    without a door.
  - go east: Agent's location: Corridor. The corridor has a closed
      sliding patio door leading north, an exit to the east without a
      door, an entranceway to the south without a door, and an
      entranceway to the west without a door.
    - go east: Agent's location: Bathroom. The bathroom contains an empty toilet. The bathroom
        has an exit to the west without a door.
    - open sliding patio door: You open sliding patio door.
      - go north: Agent's location: Backyard. The backyard contains a patio chair, a patio table,
          and a BBQ. The backyard has a closed wooden door leading
          east, an open sliding patio door leading south, and an
          entranceway to the north without a door.
  ...

- in_driveway: Agent is in the Driveway. The driveway has an open
    fiberglass door leading north and an exit to the east without a
    door.
  - go east: Agent's location: Street. The street has a closed sliding
      door leading east and an exit to the west without a door. [reach
      in_street]

- in_street: Agent is in the Street. The street has a closed sliding
    door leading east and an exit to the west without a door.
  - open sliding door: You open sliding door.
    - go east: Agent's location: Supermarket. The supermarket contains
        an empty showcase. The supermarket has an open sliding door
        leading west.
```

The Extractor then adds the two new locations to the instance context. These locations also reveal additional Unknowns to explore, which AutoContext may continue to pursue.

**Instance Context**

```
...

- Corridor:
  - objects: Nothing
```

```
    - west: entranceway (without door) to Kitchen
    - east: exit (without door) to Bathroom
    - north: closed sliding patio door to Backyard
    - south: entranceway (without door) to Bedroom

...

- Bathroom:
    - objects: empty toilet
    - west: exit (without door) to Corridor
    - east: None
    - north: None
    - south: None

- Backyard:
    - objects: patio chair, patio table, BBQ
    - west: None
    - east: closed wooden door to Unknown
    - north: entranceway (without door) to Unknown
    - south: open sliding patio door to Corridor
```

## C.2   A Trajectory of ReAct with Instance Context

In this trajectory, the ReAct agent uses the instance context provided by AutoContext to successfully navigate to the kitchen, where it locates the recipe and gathers the available ingredients. It then proceeds to the garden to collect the remaining ingredients. Afterwards, the agent processes all ingredients, prepares the meal, and eats it to win the game. Outputs of the LLM that reflect its use of the instance context are highlighted with a red background. Part of the observations and steps are omitted for brevity and clarity.

---

**A Trajectory of ReAct with the Instance Context**

Step 1:
```
[Observation]: You are hungry! Let's cook a delicious meal. Check the
    cookbook in the kitchen for the recipe. Once done, enjoy your meal!
-= Livingroom =- You are in a livingroom...

[Thought]: I need to find the cookbook in the kitchen to check the
    recipe for the meal. First, I should move from the livingroom to
    the kitchen. According to the instance context, the kitchen is
    north of the livingroom. I'll go north to reach the kitchen.

[Action]: go north
```

Step 2:
```
[Observation]: -= Kitchen =- You arrive in a kitchen. A normal kind of
    place. The room seems oddly familiar, as though it were only
    superficially different from the other rooms in the building...

[Thought]: Now that I'm in the kitchen, I need to check the cookbook on
     the table to see the recipe for the meal. The cookbook is visible
     on the table, so I'll examine it to understand what ingredients and
     cooking methods are required.

[Action]: examine cookbook
```

Step 3:
```
[Observation]: You open the copy of "Cooking: A Modern Approach (3rd Ed
    .)" and start reading:
```

---

```
Recipe #1
---------
Gather all following ingredients and follow the directions to prepare
    this tasty meal.

Ingredients:
orange bell pepper
pork chop
purple potato
red onion
white onion

Directions:
dice the orange bell pepper
grill the orange bell pepper
slice the pork chop
grill the pork chop
slice the purple potato
grill the purple potato
slice the red onion
roast the red onion
dice the white onion
grill the white onion
prepare meal

[Thought]: Based on the recipe, I need to gather and prepare the
    following ingredients:
1. orange bell pepper (needs dicing and grilling)
2. pork chop (needs slicing and grilling)
3. purple potato (needs slicing and grilling)
4. red onion (needs slicing and roasting)
5. white onion (needs dicing and grilling)

First, I should check my inventory to see what I already have. Then I'
    ll need to collect any missing ingredients from the fridge or
    garden. The knife is already on the counter, which I'll need for
    slicing/dicing.

[Action]: inventory

Step 4:
[Observation]: You are carrying nothing.

[Thought]: Since my inventory is empty, I need to collect all the
    required ingredients. Let's start by gathering what's available in
    the kitchen:
1. The fridge contains white onion, red onion, and raw pork chop
2. The counter has raw purple potato
3. The orange bell pepper is in the garden (according to instance
    context)

I'll first take the knife from the counter since I'll need it for
    slicing/dicing, then collect ingredients from the fridge.

[Action]: take knife from counter

Step 5:
[Observation]: You take the knife from the counter.

```

```
[Thought]: Now that I have the knife, I should collect the ingredients
    from the fridge. According to the recipe, I need:
- white onion
- red onion
- pork chop

These are all in the closed fridge in the kitchen. I'll open the fridge
    first to access them.

[Action]: open fridge

Step 6:
[Observation]: You open the fridge, revealing a white onion, a red
    onion, a raw pork chop, a yellow bell pepper, a block of cheese, a
    red hot pepper, a raw white tuna and a yellow onion.

[Thought]: Now that the fridge is open, I can take the required
    ingredients:
1. white onion
2. red onion
3. raw pork chop

I'll take these three items first since they're needed for the recipe.
    The other items (yellow bell pepper, cheese, etc.) aren't part of
    the recipe so I can ignore them for now.

[Action]: take white onion from fridge

Step 7:
[Observation]: You take the white onion from the fridge.

Your score has just gone up by one point.

[Thought]: Successfully got the white onion. Next I need to take the
    red onion from the fridge, which is another required ingredient for
     the recipe. After that, I'll take the pork chop.

[Action]: take red onion from fridge

Step 8:
[Observation]: You take the red onion from the fridge.

Your score has just gone up by one point.

[Thought]: Now I have both onions. Next I need to take the pork chop
    from the fridge, which is the third required ingredient currently
    available here. After this, I'll need to get the purple potato from
     the counter and the orange bell pepper from the garden.

[Action]: take raw pork chop from fridge

Step 9:
[Observation]: You take the pork chop from the fridge.
```

```
Your score has just gone up by one point.

[Thought]: Now I have the pork chop. Next, I should take the purple
    potato from the counter since it's another required ingredient that
    's available in the kitchen. After that, I'll need to go to the
    garden to get the orange bell pepper.

[Action]: take raw purple potato from counter

Step 10:
[Observation]: You take the purple potato from the counter.

Your score has just gone up by one point.

[Thought]: Now I have the purple potato. The only remaining ingredient
    is the orange bell pepper, which according to the instance context
    is in the garden. To get there, I need to: 1. Go south to
    livingroom 2. Go north to kitchen (already here). 3. Go east to
    corridor. 4. Go north to backyard. 5. Go north to garden. Let's
    start moving towards the garden to collect the orange bell pepper.

[Action]: go east

Step 11:
[Observation]: -= Corridor =- You arrive in a corridor. An ordinary one
    .     There is a closed sliding patio door leading north...

[Thought]: Now in the corridor. Next step is to go north through the
    sliding patio door to reach the backyard, which leads to the garden
     where the orange bell pepper is located. I'll open the sliding
    patio door first.

[Action]: open sliding patio door

Step 12:
[Observation]: You open sliding patio door.

[Thought]: Now that the sliding patio door is open, I can proceed north
     to the backyard. From there, I can access the garden to collect
    the orange bell pepper.

[Action]: go north

Step 13:
[Observation]: -= Backyard =- You've just sauntered into a backyard.
    You scan the room, seeing a patio chair. The patio chair is stylish
    ...

[Thought]: Now in the backyard. According to the instance context, the
    garden is north of here, and that's where the orange bell pepper is
    located. I'll proceed north to the garden to collect it.

[Action]: go north

Step 14:
```

```
[Observation]: -= Garden =- You are in a garden.     There is an exit to
    the south. Don't worry, there is no door...

[Thought]: Finally in the garden where I can see the orange bell pepper
    on the floor. This is the last ingredient needed for the recipe. I
    'll take it now to complete my ingredient collection.

[Action]: take orange bell pepper

Step 15:
[Observation]: You pick up the orange bell pepper from the ground.

Your score has just gone up by one point.

[Thought]: Now I have all the required ingredients in my inventory:
1. white onion
2. red onion
3. pork chop
4. purple potato
5. orange bell pepper

Next, I need to return to the kitchen to begin food preparation
    according to the recipe directions. The quickest way back is south
    to backyard, then south to corridor, then west to kitchen.

[Action]: go south

Step 16:
[Observation]: -= Backyard =- You've just sauntered into a backyard.
    You scan the room, seeing a patio chair. The patio chair is stylish
    ...

[Thought]: Now in the backyard. Next step is to go south through the
    open sliding patio door to return to the corridor, then west to the
     kitchen where I can start preparing the ingredients.

[Action]: go south

Step 17:
[Observation]: -= Corridor =- You arrive in a corridor. An ordinary one
    .    There is an open sliding patio door leading north...

[Thought]: Now in the corridor. The kitchen is to the west through an
    entranceway without a door. I'll go west to return to the kitchen
    where I can begin food preparation using the knife and cooking
    tools available there.

[Action]: go west

Step 18:
[Observation]: -= Kitchen =- You arrive in a kitchen. A normal kind of
    place. The room seems oddly familiar, as though it were only
    superficially different from the other rooms in the building...

[Thought]: Now back in the kitchen with all ingredients. According to
    the recipe, I need to:
1. dice orange bell pepper then grill it
```

```
2. slice pork chop then grill it
3. slice purple potato then grill it
4. slice red onion then roast it
5. dice white onion then grill it

First, I'll dice the orange bell pepper using the knife I already have.

[Action]: dice orange bell pepper with knife

Step 19:
[Observation]: You dice the orange bell pepper.

Your score has just gone up by one point.

[Thought]: Successfully diced the orange bell pepper. Next step is to
    grill it using the BBQ in the backyard. I'll need to:
1. Go east to corridor
2. Go north through sliding patio door to backyard
3. Use the BBQ there

Let's start moving towards the backyard.

[Action]: go east

Step 20:
[Observation]: -= Corridor =- You arrive in a corridor. An ordinary one
    .    There is an open sliding patio door leading north...

[Thought]: Now in the corridor. Next step is to go north through the
    open sliding patio door to reach the backyard where the BBQ is
    located.

[Action]: go north

Step 21:
[Observation]: -= Backyard =- You've just sauntered into a backyard.
    You scan the room, seeing a patio chair. The patio chair is stylish
    ...

[Thought]: Now in the backyard with access to the BBQ. According to the
     recipe, I need to grill the diced orange bell pepper. I'll use the
     BBQ here to cook it.

[Action]: cook orange bell pepper with BBQ

Step 22:
[Observation]: You grilled the orange bell pepper.

Your score has just gone up by one point.

[Thought]: Successfully grilled the orange bell pepper. Now I need to
    return to the kitchen to continue preparing the other ingredients:
1. slice pork chop then grill it
2. slice purple potato then grill it
3. slice red onion then roast it
```

```
4. dice white onion then grill it

First, I'll return to the kitchen by going south to corridor, then west
    to kitchen.

[Action]: go south

...

Step 42:
[Observation]: You grilled the white onion.

Your score has just gone up by one point.

[Thought]: Successfully grilled the white onion. Now all ingredients
    have been prepared according to the recipe:
1. orange bell pepper - diced and grilled
2. pork chop - sliced and grilled
3. purple potato - sliced and grilled
4. red onion - sliced and roasted
5. white onion - diced and grilled

Now I need to return to the kitchen to prepare the meal. I'll go south
    to corridor, then west to kitchen.

[Action]: go south

Step 43:
[Observation]: -= Corridor =- You arrive in a corridor. An ordinary one
    .    There is an open sliding patio door leading north...

[Thought]: Now in the corridor. Final step is to go west to return to
    the kitchen where I can prepare the meal using all the prepared
    ingredients.

[Action]: go west

Step 44:
[Observation]: -= Kitchen =- You arrive in a kitchen. A normal kind of
    place. The room seems oddly familiar, as though it were only
    superficially different from the other rooms in the building...

[Thought]: Now back in the kitchen with all ingredients prepared
    according to the recipe. The final step is to prepare the meal
    using all these ingredients in the kitchen.

[Action]: prepare meal

Step 45:
[Observation]: Adding the meal to your inventory.

[Thought]: Successfully prepared the meal! Now I can eat it to complete
    the task and win the game. The meal is in my inventory, so I'll
    eat it now.

[Action]: eat meal

[Observation]: You eat the meal. Not bad.
```

## C.3 A TRAJECTORY OF REACT WITHOUT INSTANCE CONTEXT

We present a trajectory in which the ReAct agent is not provided with the instance context. In this case, lacking contextual guidance, the agent first spends several steps attempting to locate the kitchen. By contrast, in the trajectory with instance context, the agent navigates directly to the kitchen. It then examines the cookbook and begins searching for the necessary ingredients to prepare a meal. However, it leaves the north of the corridor unexplored, which prevents it from finding the orange bell pepper and ultimately causes the task to fail. This episode shows the fragility of reasoning without systematic exploration. Key outputs of the LLM are highlighted with a red background. Part of the observations and steps are omitted for brevity and clarity.

---

**A Trajectory of ReAct without Instance Context**

Step 1:
```
[Observation]: You are hungry! Let's cook a delicious meal. Check the
    cookbook in the kitchen for the recipe...

[Thought]: I need to get to the kitchen to check the cookbook for the
    recipe. The living room has exits to the south (through a
    fiberglass door), east, and north. Since the kitchen is likely
    connected to one of these exits, I'll try going east first as it's
    mentioned as an exit without a door, which might lead to the
    kitchen or closer to it.

[Action]: go east
```

...

Step 8:
```
[Observation]: -= Kitchen =- You arrive in a kitchen. A normal kind of
    place...

[Thought]: Finally found the kitchen! Now I need to check the cookbook
    on the table to see the recipe. After examining the recipe, I'll
    gather the necessary ingredients and tools (like the knife and
    potatoes/apple on the counter) to prepare the meal.

[Action]: look cookbook
```

Step 9:
```
[Observation]: You open the copy of "Cooking: A Modern Approach (3rd Ed
    .)" and start reading:

Recipe #1
---------
Gather all following ingredients and follow the directions to prepare
    this tasty meal.

Ingredients:
orange bell pepper
pork chop
purple potato
red onion
white onion

Directions:
dice the orange bell pepper
grill the orange bell pepper
slice the pork chop
grill the pork chop
slice the purple potato
```

```
grill the purple potato
slice the red onion
roast the red onion
dice the white onion
grill the white onion
prepare meal

[Thought]: The recipe requires several ingredients and specific
    preparation methods. Looking around the kitchen, I see a raw purple
     potato and a knife on the counter, which are both needed for the
    recipe. I should first take these items into my inventory before
    proceeding with the preparation steps.

[Action]: take purple potato from counter

...

Step 25:
[Observation]: You take the pork chop from the fridge.

Your score has just gone up by one point.

[Thought]: Now I have the pork chop, red onion, and white onion. The
    only missing ingredient is the orange bell pepper. Since it's not
    in the fridge or on the counter, I'll check the inventory to see
    what I have and then look around other rooms for the orange bell
    pepper.

[Action]: inventory

Step 26:
[Observation]: You are carrying: a raw pork chop, a red onion, a white
    onion, a knife and a sliced grilled purple potato.

[Thought]: I have all ingredients except the orange bell pepper. Since
    it's not in the kitchen, I'll need to explore other rooms. I recall
     seeing a bedroom and bathroom earlier. I'll check the bedroom (go
    south from kitchen) first for the orange bell pepper.

[Action]: go south

Step 27:
[Observation]: -= Livingroom =- You are in a livingroom. A standard
    kind of place...

[Thought]: This is the living room, not the bedroom. I must have taken
    a wrong turn. I'll go back north to the kitchen and then try going
    east to reach the bedroom where I might find the orange bell pepper
    .

[Action]: go north

Step 28:
[Observation]: -= Kitchen =- You arrive in a kitchen. A normal kind of
    place...

[Thought]: Back in the kitchen, I'll try going east this time to reach
    the bedroom. The orange bell pepper might be there or in another
```

```
       room. Since it's not in the kitchen, I need to search other
       locations systematically.

  [Action]: go east

  Step 29:
  [Observation]: -= Corridor =- You arrive in a corridor. An ordinary one
       ...

  [Thought]: Now in the corridor, I'll try going east to check the
       bathroom (though unlikely to have food) and then south to check the
        bedroom if needed. The orange bell pepper might be in one of these
        rooms.

  [Action]: go east

  Step 30:
  [Observation]: -= Bathroom =- You arrive in a bathroom. A typical kind
       of place...

  [Thought]: The bathroom doesn't have any ingredients. I'll return to
       the corridor (go west) and then try going south to check the
       bedroom for the orange bell pepper. If it's not there, I might need
        to check other locations or consider that the yellow bell pepper
       in the fridge might be acceptable (though the recipe specifies
       orange).

  [Action]: go west

  Step 31:
  [Observation]: -= Corridor =- You arrive in a corridor. An ordinary one
       ...

  [Thought]: Now back in the corridor, I'll go south to check the bedroom
        for the orange bell pepper. If it's not there, I might need to
       reconsider the recipe requirements or check if the yellow bell
       pepper can substitute for orange (though the guidelines say colors
       must match exactly).

  [Action]: go south

  Step 32:
  [Observation]: -= Bedroom =- You've entered a bedroom.  You can make
       out a bed...

  [Thought]: The bedroom doesn't have the orange bell pepper either.
       Since I've checked all accessible rooms (kitchen, fridge, bathroom,
        bedroom) and only found a yellow bell pepper, I might need to
       proceed with the recipe steps I can complete while keeping an eye
       out for the orange bell pepper. I'll return to the kitchen (go
       north then west) to start processing the ingredients I have.

  [Action]: go north

  Step 33:
  [Observation]: -= Corridor =- You arrive in a corridor. An ordinary one
       ...
```

```
[Thought]: Now in the corridor, I'll head west to return to the kitchen
    . There, I can start processing the ingredients I have (pork chop,
    red onion, white onion) according to the recipe steps while
    continuing to look for the orange bell pepper in other areas if
    needed.

[Action]: go west

Step 34:
[Observation]: -= Kitchen =- You arrive in a kitchen. A normal kind of
    place...

[Thought]: Back in the kitchen, I'll proceed with processing the
    ingredients I have. According to the recipe, I need to:
1. Slice the pork chop (I have knife and pork chop)
2. Slice the red onion
3. Dice the white onion
I'll start with slicing the pork chop since that's a main ingredient.

[Action]: slice pork chop with knife

...

Step 46:
[Observation]: You grilled the white onion.

Your score has just gone up by one point.

[Thought]: The white onion has been successfully grilled. Now all that
    remains missing is the orange bell pepper to complete the recipe.
    Since I couldn't find it earlier, I'll check the fridge again in
    case I missed it (go south to corridor, then west to kitchen).

[Action]: go south

Step 47:
[Observation]: -= Corridor =- You arrive in a corridor. An ordinary one
    ...

[Thought]: Now in the corridor, I'll go west to return to the kitchen
    and check the fridge again for the orange bell pepper. If it's
    still not there, I might need to explore other rooms more
    thoroughly or consider that the yellow bell pepper might be
    acceptable despite the recipe specifying orange.

[Action]: go west

Step 48:
[Observation]: -= Kitchen =- You arrive in a kitchen. A normal kind of
    place...

[Thought]: The fridge still only has a yellow bell pepper, not orange.
    According to the guidelines, ingredients must exactly match the
    color in the recipe. Since I can't find an orange bell pepper, I'll
     check the cookbook again to confirm the requirements and see if
    there might be another location for it.
```

```
[Action]: look cookbook

Step 49:
[Observation]: You open the copy of "Cooking: A Modern Approach (3rd Ed
    .)" and start reading:

Recipe #1
---------
Gather all following ingredients and follow the directions to prepare
    this tasty meal.

Ingredients:
orange bell pepper
pork chop
purple potato
red onion
white onion

Directions:
dice the orange bell pepper
grill the orange bell pepper
slice the pork chop
grill the pork chop
slice the purple potato
grill the purple potato
slice the red onion
roast the red onion
dice the white onion
grill the white onion
prepare meal

[Thought]: The recipe clearly specifies an orange bell pepper, and the
    fridge only has a yellow one. Since I've searched all accessible
    rooms (kitchen, fridge, bathroom, bedroom, backyard) without
    finding an orange bell pepper, I might need to check the closed
    frosted-glass door leading north in the kitchen as a last resort
    before concluding the orange bell pepper is unavailable.

[Action]: open frosted-glass door

Step 50:
[Observation]: You open frosted-glass door.

[Thought]: The frosted-glass door is now open. I'll go north through it
     to check if there's another room that might contain the orange
    bell pepper. This is my last hope to find the missing ingredient.

[Action]: go north

Step 51:
[Observation]: -= Pantry =- You find yourself in a pantry. A typical
    kind of place...

[Thought]: The pantry is empty and doesn't contain the orange bell
    pepper. Since I've now searched every accessible location without
    finding it, I'll have to conclude that the orange bell pepper isn't
     available in this game version. I'll return to the kitchen (go
    south) and consider preparing the meal with the ingredients I have,
     though it might not be complete according to the recipe.
```

```
[Action]: go south

Step 52:
[Observation]: -= Kitchen =- You arrive in a kitchen. A normal kind of
    place...

[Thought]: Since I can't find the orange bell pepper anywhere, I'll
    attempt to use the yellow bell pepper from the fridge as a
    substitute, despite the recipe specifying orange. I'll take it from
     the fridge and proceed with the preparation steps (dicing and
    grilling) to see if the game accepts it.

[Action]: take yellow bell pepper from fridge

Step 53:
[Observation]: You take the yellow bell pepper from the fridge.

[Thought]: Now that I have the yellow bell pepper, I'll proceed to dice
     it as the first step in its preparation according to the recipe (
    though it specifies orange bell pepper). I have the knife in my
    inventory to perform this action.

[Action]: dice yellow bell pepper with knife

...

Step 59:
[Observation]: -= Kitchen =- You arrive in a kitchen. A normal kind of
    place...

[Thought]: Now back in the kitchen, I'll attempt to prepare the meal
    with all the processed ingredients I have: grilled purple potato,
    grilled pork chop, roasted red onion, grilled white onion, and
    grilled yellow bell pepper. Even though the bell pepper color doesn
    't match exactly, it's worth trying to see if the game accepts it.

[Action]: prepare meal

Step 60:
[Observation]: You still miss something.

[Thought]: The game indicates I'm still missing something, likely
    because the yellow bell pepper doesn't match the required orange
    one in the recipe. Since I've thoroughly searched all locations and
     can't find an orange bell pepper, I'll check my inventory to see
    what processed ingredients I have and verify against the recipe
    requirements.

[Action]: inventory

Step 61:
[Observation]: You are carrying: a diced grilled yellow bell pepper, a
    sliced grilled pork chop, a sliced roasted red onion, a diced
    grilled white onion, a knife and a sliced grilled purple potato.

[Thought]: I have all required ingredients processed according to the
    recipe, except the bell pepper color is yellow instead of orange.
    Since I can't find an orange bell pepper anywhere, I'll try one
```

```
        last time to check if there's any hidden location I might have
        missed by examining the kitchen more thoroughly.

    [Action]: look counter

    Step 62:
    [Observation]: The counter is solidly built.

    On the counter are a red apple and a raw yellow potato.

    [Thought]: The counter only has a red apple and raw yellow potato –
        nothing useful for the recipe. Since I've exhausted all possible
        locations and the game won't accept the meal with the yellow bell
        pepper, I'll conclude that this version of the game might be
        unwinnable due to the missing orange bell pepper. I'll end the game
         here.

    [Action]: quit
```

## D  PROMPTS

We provide the prompt templates for the TextWorld environment here. Other environments use the same template, with only minor modifications to the examples and background information specific to each environment.

### D.1  PROMPTS FOR THE PLANNER

**Planner: Propose TODO for Observations**

```
    ## Objective

    You are assisting an agent that operates in an interactive
        environment to gather observations. Your task is to propose a new
         TODO for the agent to gather more observations for the knowledge
         document.

    ## Guidelines

    - Provide the new TODO by proposing an action sequence. The action
        sequence can start from any available state in the TODO forest.
    - All actions should be immediately executable without placeholders
        or undefined variables.
    - If the `Observation` section in the knowledge document seems
        complete, i.e., no missing entries or unknown information, you
        output `None` for the action sequence.
    - Ensure your new TODO is different from the existing TODOs in the
        forest.
    - The length of your action sequence should not exceed {{ max_length
        }}.

    ## Your workflow

    1. Analyze the current knowledge document to list unknown
        observations and missing entries in the `Observations` section.
    2. Analyze the current TODO forest to find new action sequences that
        can gather the unknown observations and missing entries.

    ## Background
```

```
{{ background }}

{% if trajectory %}
## Recent trajectory

This is the recent trajectory of the agent in the environment for
    your reference.

{{ trajectory }}

{% endif %}
## Definition of TODO forest

{{ todo_def }}

## Current TODO forest

{{ todo_forest }}

## Format of knowledge document

{{ knowledge_format }}

## Knowledge document

{{ knowledge }}

## Output format

First, analyze step by step.

Then provide your new TODO by strictly following the format below.

<thought>
You analyze step by step here.
</thought>
<missing_observations>
List of missing observations or unknown entries in the `Observations`
    section here.
</missing_observations>
<todo>
state_name -> action -> ... -> action
</todo>

## Example output

<thought>
The knowledge document requires location information in the `
    Observations` section. It contains some missing locations,
    including:
- go east and go north from the location of the init_state.
- west of the kitchen.
None of them is present in the `Observations` section and the TODO
    forest. I can choose any of them to propose a new TODO.
</thought>
<missing_observations>
- go east and go north from the location of the init_state.
- west of the kitchen.
</missing_observations>
<todo>
init_state -> go east -> go north
</todo>
```

```
## Example output

<thought>
The knowledge document requires object information in the `
    Observations` section. It seems that the knowledge document is
    already completed, i.e., all objects have been explored. I will
    not propose any new TODOs.
</thought>
<missing_observations>
Nothing is missing.
</missing_observations>
<todo>
None
</todo>
```

**Planner: Propose TODO for Action Rules**

```
## Objective

You are assisting an agent that operates in an interactive
    environment to gather action rules. Your task is to propose
    additional TODOs for the agent based on an existing TODO forest,
    by outputting a list of new TODOs.

## Guidelines

- You propose TODOs to discover the correct syntax and requirements
    for available actions.
- Ensure all proposed actions are immediately executable without
    placeholders or undefined variables.
- Some actions have preconditions. You may create a sequence of
    actions to satisfy the preconditions before the final action.
- Be creative, if an action failed,
  - Try to use it with a different preconditioning action sequence.
  - Try to use other actions that have not been tried yet.
  - Try to use different syntax or names for the objects.
    - For example, "take red carrot", "take carrot", "take red carrot
        from table", ...
    - For example, "open door", "open front door", ...
- The length of each action sequence should not exceed {{ max_length
    }}.

## Your workflow

- Find actions that have not been tried yet in the current TODO
    forest or all the results are `action failed`.
- Analyze step by step to find why the action failed, and find a
    different action sequence that could make the action succeed.
- Propose the action sequence as a new TODO.
- Propose at most {{ num_todo }} new TODOs.

{% if trajectory %}
## Recent trajectory

This is the recent trajectory of the agent in the environment for
    your reference.

{{ trajectory }}

{% endif %}
## Definition of TODO forest
```

```
{{ todo_def }}

## Background

{{ background }}

## Current TODO forest

{{ todo_forest }}

## Output format

First, analyze step by step.

Then provide your new TODOs by providing paths from any available
    state to the new TODOs, in the format below. All key results
    should be omitted for brevity. One path for each new TODO.

```json
[
  "state -> action -> action -> ... -> action",
  "state -> action -> action -> ... -> action",
  ...
]
```

## Example output

All nodes in the current TODO forest are `action failed`, but `go to`
    is not tried yet. I would like to try `go to` to see if it can
    succeed.

```json
[
  "init_state -> go to door",
  "init_state -> go to light switch"
]
```

## Example output

Previous `examine` actions all failed, but maybe I can try `examine`
    after going to the door. This is worth trying and sounds
    promising. Also, since `drive car` does not work, maybe I can try
    `use car`.

```json
[
  "init_state -> go to door -> examine door",
  "init_state -> use car"
]
```

## Example output

The current TODO forest shows that `add oil` is a precondition for `
    drive car`. As I want to try `drive car`, I will start with the
    state `added_oil`.

```json
[
  "added_oil -> drive car",
  "added_oil -> use car"
```
```

```
    ]
    ```
```

## Planner: Promote Nodes to States

```
## Objective

You create a new state for the TODO forest.

## Your workflow

1. Analyze the knowledge document to list all missing entries and
   unknown observations.
2. Find a TODO tree and one of its nodes that can help gather more
   knowledge for the knowledge document.
3. Output the new state by outputting the path from the root of the
   TODO tree to the selected node.

## Guidelines

- Prioritize new states that can help address 'Unknown' entries in
  the knowledge document, i.e., the agent can take only a few
  actions from the new state to gather the missing observations.
- The key result of the selected node should not be 'action failed'.
- The agent should be able to gather more knowledge by continuing
  exploration from the new state.
- The new state should be significantly different to all existing
  states.
- You may choose any existing state in the TODO forest as the
  starting point for your path. The ending node of the path will
  create a new state for the TODO forest.
- Do not add additional actions to the path. Your path should be
  already in the TODO forest.

## Background

{{ background }}

## Definition of TODO forest

{{ todo_def }}

## Current TODO forest

{{ todo_forest }}

## Knowledge document format

{{ knowledge_format }}

## Knowledge document

{{ knowledge }}

## Output format

First, analyze step by step.

Then, provide your response by strictly following the format below.

```json
{
```

```
      "target_missing_observation": "the missing observation or unknown
          entry in the knowledge document that you want to address",
      "selected_path": "existing_state -> action -> action -> ... ->
          action",
      "new_state_name": "descriptive name for the new state",
      "state_summary": "self-contained and brief summary of what
          characterizes this new state. Focus on facts. No assumptions or
           plans here."
}
```

## Example output

Looking at the knowledge document format, it needs location
    information. The kitchen is explored in the TODO forest and has
    some exits. The agent can continue exploring more locations from
    the kitchen. Also, currently there is no state for the kitchen.
    So I can add this state.

Looking carefully at the TODO forest, the agent enters the kitchen by
    starting from the `woke_up` state. So I will use this path to
    create the new state.

```json
{
  "target_missing_observation": "kitchen's neighboring locations",
  "selected_path": "woke_up -> open door -> enter room",
  "new_state_name": "in_kitchen",
  "state_summary": "in kitchen."
}
```

## Example output

Looking at the knowledge document format, it needs location
    information and object information. The TODO forest shows that
    arriving in a place will directly reveal the objects in that
    place. So I just need to find a place that has not been explored
    yet. The living room seems to be a good candidate, as many of its
     exits are marked as unknown in the knowledge document.

Looking carefully at the TODO forest, the agent arrives in the living
    room by starting from the `in_kitchen` state. So I will use this
    path to create the new state.

```json
{
  "target_missing_observation": "living room's neighboring locations,
      and the objects in those locations",
  "selected_path": "in_kitchen -> take a rest -> go east",
  "new_state_name": "in_living_room",
  "state_summary": "in living room."
}
```

D.2  PROMPTS FOR THE ACTOR

Below is the prompt for the subagent in the agent mode of AutoContext.

**Actor: Subagent**

```
## Objective

You control an agent in an interactive environment. The agent can
    perform various actions in the environment. Each action will
    return a result as a string.

## Guidelines

- Strategic Planning: Plan your actions strategically to efficiently
    complete the task, but remain flexible to pivot when new
    information emerges.
- Adaptive Learning: Pay attention to your recent action results and
    adapt your strategy accordingly.

## Background

{{ background }}

## Output format

Provide your response by strictly following the format below. Note
    that you can output only one action.

<thought>
Analyze step by step here.
</thought>
<action>
Your action here
</action>
```

## D.3  PROMPTS FOR THE EXTRACTOR

**Extractor: Propose Edits for Observations**

```
## Objective

You are an expert in analyzing LLM agent's trajectory.

An agent is operating in an interactive environment. You will be
    given the trajectory of the agent, and a knowledge document about
     the environment.

Your task is to analyze the trajectory step by step, and modify the `
    Observation` section in the knowledge document accordingly.

If no modification is needed, output `None` for your modification.

## Background

{{ background }}

## Guidelines

- Find objects that are observed in the trajectory. Add them to the
    knowledge document if they are not already there.
- Only write the required properties in the knowledge document.
- Output your knowledge modification items
  - add
  - update: from ... to ...
  - remove
```

```
- Correct the errors in the knowledge document with 'update' if you
    find any.
- The knowledge document was built by previous trajectories. Use the
    current trajectory to add knowledge and correct errors, but do
    not remove any existing knowledge.

## Trajectory

{{ trajectory }}

## Definition of knowledge

{{ knowledge_definition }}

## Current knowledge

{{ knowledge }}

## Output format

First, analyze step by step.

Then, output your decision by strictly following the format below.

<thought>
Your analysis here.
</thought>
<modification1>
Introduce how the knowledge should be modified here. / None
</modification1>
<modification2>
Introduce how the knowledge should be modified here. / None
</modification2>
...
```

**Extractor: Propose Edits for Action Rules**

```
## Objective

You are an expert in analyzing LLM agent's trajectory.

An agent is operating in an interactive environment. You will be
    given the agent's trajectory and a knowledge document about the
    environment.

Your task is to analyze the agent's trajectory step by step, and
    modify the 'Action Rules' section in the current knowledge
    document accordingly.

If no modification is needed, output <modification1>None</
    modification1>.

## Background

{{ background }}

## Definition of knowledge

{{ knowledge_definition }}

## Guidelines
```

```
- List all successful actions taken by the agent in the trajectory.
- Check if the successful actions are already in the knowledge
    document. If not, add them to the knowledge document.
- Analyze the observations before and after the action carefully to
    identify the requirements, the key results, and the key
    observations.
- Double check the requirements to make sure they are sufficient to
    achieve the key results.
- Output your knowledge modification items
  - add
  - update: from ... to ...
  - remove
- The knowledge document was built by previous trajectories. Use the
    current trajectory to add knowledge and correct errors. Do not
    remove any knowledge unless you have enough evidence to show that
     the existing knowledge is incorrect.
- Do not modify the `Observations` section. Only modify the `Action
    Rules` section.

## Agent's trajectory

Below is the recent trajectory of the agent's actions in the
    environment. Earlier actions have been omitted for brevity.

{{ trajectory }}

## Current knowledge

{{ knowledge }}

## Output format

First, analyze step by step.

Then, provide your response by strictly following the format below.

<thought>
Your analysis here.
</thought>
<modification1>
Introduce how the knowledge should be modified here.
</modification1>
<modification2>
Introduce how the knowledge should be modified here.
</modification2>
...

## Example modifications

<modification1>
Add:
- Action: Make Paper Box
  - Requirements: 1 scissor, 1 paper
  - Key Result: obtain paper box.
  - Note: None
</modification1>

## Example modifications

<modification1>
None
</modification1>
```

**Extractor: Check Edits**

```
## Objective

You are an expert in analyzing LLM agent's trajectory and knowledge.

You will be given
- a trajectory of an agent in an interactive environment.
- current knowledge about the environment
- a modification that someone wants to make to the current knowledge

Your task is to check if the modification is correct, and if not,
    provide the correct modifications.

## Background

{{ background }}

## Definition of knowledge

{{ knowledge_definition }}

## Guidelines

- Check if the modification is correct based on the trajectory.
- If the modification is correct, output 'Accept'. (No need to be
    very strict. As long as the modification seems to be reasonable
    and is consistent with the trajectory, you accept it or revise it
    )
- If the modification has some errors and you have enough information
     to correct it, output 'Revise', and correct it based on the
    trajectory.
- If the modification is incorrect and cannot be corrected, output '
    Reject'.
- Revise knowledge that indicates something cannot be interacted with
    . Simply remove all information that indicates something cannot
    be interacted with. For example,
  - "door (cannot be opened)" should be "door"
  - "The 'take' action cannot take the carrot" should be removed
  - "The door cannot be opened" should be removed
- Make sure the knowledge is well-supported.

## Current knowledge

{{ knowledge }}

## Trajectory

{{ trajectory }}

## Modification

{{ modification }}

## Output format

First, analyze step by step.

Then, output your decision by strictly following the format below.

<thought>
Your analysis here.
</thought>
<decision>
```

```
Accept/Revise/Reject
</decision>
<content>
If the decision is 'Revise', provide the corrected modification here.
    If the decision is 'Reject' or 'Accept', provide 'None'.
</content>
```

**Extractor: Apply Edits**

```
## Objective

You will be given
- a knowledge document about an interactive environment.
- a list of modifications that should be made to the knowledge
    document.

Your task is to apply the modifications to the knowledge document.

You output the modified knowledge document, which should preserve all
    important details and be well-organized.

## Definition of knowledge

{{ knowledge_definition }}

## Guidelines

- Remove duplicate or repetitive knowledge that conveys the same
    meaning.
- Write knowledge strictly following the format in 'Definition of
    knowledge'.
- Remove anything that doesn't follow the format in 'Definition of
    Knowledge'.

## Knowledge

{{ knowledge }}

## Modification

{{ modification_list }}

## Output format

Provide your response by strictly following the format below.
<thought>
You analyze step by step here.
</thought>
<knowledge>
Your organized and structured knowledge here. Make sure to preserve
    all important details. Do not use complex formatting. For example
    , do not use ** to emphasize words. Avoid redundancy.
</knowledge>
```

## D.4 PROMPTS FOR ENVIRONMENTS

Below are the prompts provided in the background field of the prompt templates for our main experiments.

---

**Background of TextWorld**

```
#### Available Actions

Available actions include but are not limited to:
- look: describe the current room
- look ...:  describe a specific object in the room
- inventory: print player's inventory
- go ...:    move the player north, east, south or west
- examine ...:   examine something more closely
- eat ...:   eat edible food
- open ...:  open a door or a container
- close ...: close a door or a container
- drop ...:  drop an object on the floor
- take ...:  take an object that is on the floor
- put ... on ...: place an object on a supporter
- take ... from ...:   take an object from a container or a supporter
- insert ... into ...: place an object into a container
- lock ... with ...:   lock a door or a container with a key
- unlock ... with ...: unlock a door or a container with a key
- prepare meal:  prepare a meal using ingredients in the inventory.
    You can only prepare meals in the Kitchen.

#### Tips
- No door is locked. All doors can be opened, even if it appears to
    be obstructed. For example, "open front door".
- You can examine the cookbook to see the recipe when it is visible.
- The BBQ is for grilling things, the stove is for frying things, and
     the oven is for roasting things. Cooking ingredients in the
    wrong way will lead to a failure of the game.
- Once you have processed ingredients and the appropriate cooking
    tool ready, cook all of them according to the recipe.
- There are two conditions to correctly cook something (grill/fry/
    roast): a) the ingredient you want to cook is in your inventory
    and b) there is a suitable cooking tool in the room, and then use
     'cook ... with ...' command.
- When you need to chop/slice/dice ingredients, you need to take the
    knife and the ingredient in your inventory and then 'slice/chop/
    dice ... with knife'
- Make sure to first process the food (chop/slice/dice) before you
    try to cook it.
- When you have all the ingredients (that got processed or cooked
    according to the recipe), you can 'prepare meal' in the kitchen
    and then 'eat meal' to win the game.
- The ingredients should EXACTLY match the color in the recipe, but
    if the recipe doesn't specify color, any color would be fine.
    When you 'take ... with ...', use the EXACT name you see.
- You don't need to examine the container/supporter (e.g. toolbox)
    when it says something like "there isn't a thing on it"/"has
    nothing on it"
```

---

**Background of ALFWorld**

```
If an action failed, the observation will be 'Nothing happens'.

#### Available Actions

- go to [object]
- open [object]
- close [object]
- take [object] from [object]
- put [object] in_on [object]
```

---

```
- heat [object] with [object]
- cool [object] with [object]
- clean [object] with [object]
- inventory
- look
- use [object]
- examine [object]

#### Tips

- First, use `go to` to reach the object. Then you can interact with
    it.
```

**Background of Crafter**

```
The agent is in a 2D grid world, where it can move around, interact
    with objects, and perform various actions. Each position is
    represented as [x, y], where x increases eastward, y increases
    southward. All distances are measured by Manhattan distance, i.e.
     the summation of x distance and y distance.

#### Available Actions

- Move To [x, y]
- Move West
- Move East
- Move North
- Move South
- Do
- Sleep
- Noop
- Place Stone
- Place Table
- Place Furnace
- Place Plant
- Make Wood Pickaxe
- Make Wood Sword
- Make Stone Pickaxe
- Make Stone Sword
- Make Iron Pickaxe
- Make Iron Sword

#### Tips

- Some actions may need to do multiple times to obtain the final
    effect.
- Some items may need multiple materials to craft.
- Achievements will be unlocked when they are completed for the first
    time.
- Check if resources appear in your recent observation, if you see
    them and need them, try to collect them.
```

## E  MORE RELATED WORK

**World Models for LLM Agents.**    Integrating world models into the reasoning loop of LLM-based agents is an emerging direction. DreamerV2 (Hafner et al.) learns a discrete latent dynamics model and achieves human-level Atari performance by planning in its learned state space. Hao et al. (2023) argue that effective chain-of-thought reasoning in an LLM agent can be viewed as implicit world-model planning, and propose techniques to align the LLM's reasoning with a latent world dynamics model. Chae et al. (2025) take a more direct approach by training a separate world-model module

that simulates the outcome of the agent's actions in a web navigation task. These world-model methods emphasize predicting environment dynamics or outcomes, whereas our work focuses on extracting static but critical instance facts.

**LLM Agent Frameworks.** Various agent frameworks and techniques introduce architectural improvements to better coordinate an LLM's reasoning and acting (Yao et al., 2023b;a; Lin et al., 2023; Prasad et al., 2024; Yang et al., 2024; Xiong et al., 2024; Zhu & Simmons, 2024; Wang et al., 2024b; Schick et al., 2023; Brohan et al., 2023; Zhao et al., 2024b). ReAct (Yao et al., 2023b) uses chain-of-thought reasoning steps with actions to enable more coherent and informed decisions. SwiftSage (Lin et al., 2023) expands on this idea by combining fast reactive thinking for straight-forward steps with slow deliberative planning for more complex decisions. ADaPT (Prasad et al., 2024) proposes an on-demand task decomposition planner. It attempts high-level plans but if the agent gets stuck on a subtask, the method recursively breaks that subtask down further, dynamically adjusting the plan hierarchy to the LLM's capabilities and the task complexity. Our AutoContext approach is orthogonal to these agent architectures. Rather than altering how an agent plans or executes, we provide a plug-in knowledge document that any of these agents can leverage to boost their performance in a new instance.

**Open-ended World Agent.** There exists a line of work (Wang et al., 2024a; Zhu et al., 2023; Wang et al., 2024c) on developing capable agents in open-ended environments such as Minecraft. Ghost in the Minecraft (GITM) (Zhu et al., 2023) introduces a hierarchical framework that integrates large language models with text-based knowledge and memory to decompose long-horizon goals into structured actions. JARVIS-1 (Wang et al., 2024c) couples a multimodal language model with a memory mechanism, enabling self-improvement through lifelong learning.

