# OpenReview forum: "Beyond Manuals and Tasks: Instance-Level Context Learning for LLM Agents"
_ICLR.cc/2026/Conference — ICLR 2026 Conference Withdrawn Submission_

### Official Review · Reviewer_FSMA · 2025-11-01

**Soundness:** 3
**Presentation:** 3
**Contribution:** 2
**Rating:** 6
**Confidence:** 4

**Summary:**

The authors propose AutoContext for instance-level context learning: a one-off guided exploration that builds a hand-designed instance schema (facts like locations, preconditions, recipes) and then writes a reusable context document to condition downstream agents. They evaluate on TextWorld, ALFWorld, and Crafter and report large gains when appending the learned instance document to standard agents.

**Strengths:**

* Clear problem framing; easy to follow.
* Important direction: reliable experience-derived knowledge for LLM agents.
* Formalization of ILCL and a tidy plan–act–extract framework with a compact exploration representation (their “TODO forest”).

**Weaknesses:**

* Missing baselines/comparisons. No direct comparison to KnowAgent (NAACL’25), and limited positioning vs. Tree-of-Thought [1] and RAP [2]. These are natural points of contrast for knowledge-/search-augmented agents.
* One-off exploration assumptions. Paper does not convincingly handle (i) exploration errors (incorrect facts recorded once then reused) and (ii) non-stationarity (environments that change—e.g., items moved after exploration). A mechanism for staleness detection and invalidation of facts seems required.
* Reporting gaps. Many results lack uncertainty bars. Please add error bars / CIs (or at least σ over seeds) for Table 2, Fig. 4, Fig. 5, Table 3, Table 4 for consistency.
* Manual schema design. Each domain needs a bespoke instance schema; scalability is unclear. The paper mentions this as a limitation but does not demonstrate even a minimal attempt at schema induction or robustness to schema misspecification.

**Questions:**

* How do you detect and correct wrong facts written during exploration? How do you expire/update facts in non-stationary instances?
* Can the exploring agent incorporate heuristic frontiering (e.g., A*/best-first over knowledge gaps) to improve coverage efficiency in long-horizon instances?
* Please provide an ablation with more Reflexion trials (beyond 3) and comment on trade-offs vs. AutoContext.
* Did you try any automatic schema discovery (prompted slot induction, bootstrapping from trajectories)? Any preliminary results?
* (Minor but central to the amortization claim) Can you show reuse across multiple different tasks in the same instance to demonstrate the multi-task benefit concretely?

# References
* [1] Yao et al., Tree of Thoughts (2024), arXiv:2305.10601.
* [2] Kagaya et al., RAP: Retrieval-Augmented Planning with Contextual Memory (NeurIPS’24 Wksp).
* [3] Zhu et al., KnowAgent: Knowledge-Augmented Planning for LLM-based Agents (NAACL’25).

---

### Official Review · Reviewer_fLXX · 2025-11-01

**Soundness:** 2
**Presentation:** 2
**Contribution:** 2
**Rating:** 2
**Confidence:** 4

**Summary:**

This paper presents **AutoContext**, a framework that solves agent tasks by constructing instance-level context and planning/doing actions according to it. The framework has two main components: a TODO forest and a plan-act-extract loop. The TODO forest is composed of tasks organized as a list of trees. The instance-level context contains information specific to the instance(and the environment). The plan-act-extract loop functions as the main loop that utilizes the TODO forest and the instant-level context: the planner produces TODOs, the actor completes them, and the extractor updates the context according to environment feedback, and the planner updates the TODO forest before proposing both TODOs .  The paper tests the methods on two benchmarks, TextWorld and ALFWorld, and results show that adding the component to existing frameworks yields better results on these benchmarks.

From my understanding, the main contribution of the paper is to organize the memory with the TODO forest and the context schema.

**Strengths:**

The paper targets an important problem, which is to solve partial-observable agent problems with memory and the utilization of the memory.

The idea of managing context dynamically, though may not novel, is still interesting to the community.

The method is a plug-in module that can be added to existing framework for better performance.

**Weaknesses:**

- The manuscript is confusing when trying to explain some of its core ideas. For example, section 4.2 attempts to explain the TODO forest, but the explanation of action and agent mode seems weird to me. It does not explain clearly (1) what is being stored in each case (2) how the information is stored. (3) What the exact criteria is to select the method.

- The main concern I have is the generalization and scaling ability of the context schema and the TODO List.  While the context schema appears compact, I believe it is hand-crafted and highly fine-grained, which means that the system requires very detailed knowledge about this environment before it can get started. This weakens the potential of the framework to generalize to unseen environments. Evidence This is shown in figure 3. The schema seems to be designed specifically for ALFWorld. Moreover, it has the following attribute: "- [object]:  **has_in_or_on**: [object], [object], ..." As I've also worked on this benchmark before, I think this is because ALFWorld accepts actions in a very specific syntax that has to write "put [obj] in/on [place]"(the in/on needs to appear together). If this is auto-discovered by the framework, then I believe my concern is addressed; but if it is pre-defined, I wonder how the framework could develop a good schema without extensive prior knowledge of the environment. Moreover, the chosen benchmarks are relatively simple: ALFWorld is almost solved by SOTA agents, which can also be shown in the setup Reflexion + GPT 4.1. The community has developed more complex, realistic benchmark such as webshop and webarena. These tasks are more complex and the schema is, I believe, much more difficult to pre-define/figure out.

- Insufficient Experiments. This is just to follow the last point that we may expect more experiments on more complex environments such as webshop and webarena.

**Questions:**

See weaknesses for main concerns. Below are the ones with high priority:

- Are the context schema and the TODO-Forest auto-discovered or predefined?
- Could you explain the action mode and agent mode more clearly in section 4.2?
- For the TODO-Forest, what exactly is stored in each node?
- Could you evaluate the method on more complex experiments such as webshop and webarena?

---

### Official Review · Reviewer_ASds · 2025-11-01

**Soundness:** 2
**Presentation:** 3
**Contribution:** 2
**Rating:** 4
**Confidence:** 3

**Summary:**

This paper introduces AutoContext framework, which enable LLM agents to automatically discover and record reusable, instance-specific knowledge as contexts. By structuring exploration through a “TODO forest” and validating extracted facts into a document, AutoContext improves agent performance on TEXTWORLD, ALFWORLD, and CRAFTER.

**Strengths:**

- The paper focuses on a third category of context: instance-level context, which is an interesting topic.
- The method is tested with several models and agent architectures, demonstrating consistent improvements.

**Weaknesses:**

- The evaluation is confined to a few environments with a small number of static environment instances. The approach has not been shown to scale to open-ended, dynamic settings such as web, mobile, or everyday life scenarios, where environment layouts change frequently. In such real-world cases, the assumption that a single static document D_e can be reused across tasks is unrealistic for many cases of the instance-level background.
- The planner assumes access to environment state replay or checkpointing so it can “return” to a prior node. This is rarely available outside controlled simulators. If that assumption holds, comparisons should include tree-search or hierarchical planning agents that also maintain explicit state graphs[1].
- While the authors acknowledge that the method is limited by the LLM’s context window and defer handling large observations to future work, the overall pipeline of exploring, storing structured knowledge, and reusing it at test time is very similar to many existing approaches in realistic environments, especially those in web and mobile agents that summarized application documentation and reused it through retrieval-augmented or structured-memory mechanisms (for example, [2, 3]). In addition, the way AutoContext incorporates instance-level context is closely related to earlier research on storing state information or building task-specific world models, which are not cited (for example, [4, 5]).

---
[1] Zhou, Andy, et al. "Language agent tree search unifies reasoning acting and planning in language models." arXiv preprint arXiv:2310.04406 (2023).

[2] Zhang, Chi, et al. "Appagent: Multimodal agents as smartphone users." Proceedings of the 2025 CHI Conference on Human Factors in Computing Systems. 2025.

[3] Sun, Yuchen, et al. "Gui-xplore: Empowering generalizable gui agents with one exploration." Proceedings of the Computer Vision and Pattern Recognition Conference. 2025.

[4] Rozanov, Nikolai, and Marek Rei. "Stateact: Enhancing llm base agents via self-prompting and state-tracking." arXiv preprint arXiv:2410.02810 (2024).

[5] Tang, Hao, Darren Key, and Kevin Ellis. "Worldcoder, a model-based llm agent: Building world models by writing code and interacting with the environment." Advances in Neural Information Processing Systems 37 (2024): 70148-70212.

**Questions:**

- Are the environment steps used during the AutoContext exploration phase included in the reported step budgets for downstream agents? It seems weird that 10 steps is enough for alfworld tasks.

---

### Official Review · Reviewer_1V1p · 2025-11-01

**Soundness:** 3
**Presentation:** 3
**Contribution:** 2
**Rating:** 2
**Confidence:** 3

**Summary:**

The paper defines Instance-Level Context Learning and presents AutoContext, a method that performs guided exploration of a new environment to produce an agent-readable document of reusable facts (locations, objects, preconditions, action rules). AutoContext is organized around a schema with explicit Unknown markers that expose knowledge gaps and a TODO forest that structures exploration; a plan–act–extract loop repeatedly proposes TODOs, executes them, and updates the facts document. Experiments on TextWorld, ALFWorld, Crafter show large gains when simply appending the facts document in-context to diverse agents, and ablations attribute separate gains from the planner, extractor, and TODO forest.

**Strengths:**

* **Plug-and-play benefits across agents.** The facts document is appended to ReAct/IGE/Reflexion with strong, consistent improvements in success and sample efficiency; ablations isolate the role of each component.
* **Clarity and reproducibility.** The paper is generally well written, with helpful figures, detailed appendices (including schema examples and prompts), and a clear experimental narrative.

**Weaknesses:**

* **Novelty is incremental relative to instance-memory / reward-free exploration.** The ILCL formalization reads as an immediate generalization of existing LLM fact-augmentation methods \[1\], which, by itself, cannot be considered novel, since it is commonplace outside LLMs under the name of reward-free exploration. From this perspective, the first claimed contribution appears overstated.
* **The schema appears environment-specific / engineered.** Figure 3 and the Appendix suggest substantial manual schema design per domain (navigation slots, container relations, crafting rules). This undercuts the critique that prior methods are “ad-hoc prompts,” since the schema itself encodes a bias toward selected fact types. The second claim of being a “task-agnostic method” is also unclear.
* **Assumptions about state control and determinism.** Resuming from TODO nodes relies on replaying trajectories, which may not be possible in general; moreover, the method seems tailored to deterministic MDPs and might not apply out of the box to stochastic settings. These assumptions/limitations should be explicitly stated early in the paper and also undermine the “task-agnostic” claim.
* **Positioning vs. “biased/partial” prior memories.** The paper argues that, in online fact generation \[1\], prior memories are task-biased, but AutoContext’s schema also biases what is recorded. Additionally, for the tasks evaluated, online fact generation would be highly applicable, and a comparison with such a method would be required to justify the paper’s claims. Without comparisons to task-driven memory baselines on the same tasks, it’s hard to justify the claimed benefit.

\[1\] Holt, S., Luyten, M. R., & Pouplin, T. (2025). *Improving LLM Agent Planning with In-Context Learning via Atomic Fact Augmentation and Lookahead Search.* arXiv:2506.09171.

**Questions:**

1. **Environment control & stochasticity.** Exactly what guarantees are assumed for reset/save/restore and transition determinism? How would AutoContext behave if replay to TODO nodes is unreliable, or if resets are expensive?
2. **Schema sensitivity.** Could we remove the schema altogether for all components (not just for the Extractor, as done in the ablation)?

---

### Note · Authors · 2025-12-03

I have read and agree with the venue's withdrawal policy on behalf of myself and my co-authors.